# Inconsistency-Aware Minimization: Improving Generalization with Unlabeled Data

**Hee-Sung Kim** [1]  **Hyeonseong Kim** [1 2]  **Sungyoon Lee** [1]

## Abstract

Estimating the generalization gap and developing optimization methods that improve generalization are crucial for deep learning models, for both theoretical understanding and practical applications. Leveraging unlabeled data for these purposes offers significant advantages in real-world scenarios. This paper introduces a novel generalization measure, *local inconsistency*, derived from an information-geometric perspective on the parameter space of neural networks. A key feature of local inconsistency is that it can be computed without explicit labels. We establish theoretical underpinnings by connecting local inconsistency to the Fisher information matrix and the loss Hessian. Empirically, we demonstrate that local inconsistency correlates with the generalization gap. Based on these findings, we propose Inconsistency-Aware Minimization (IAM), which incorporates local inconsistency into the training objective. We demonstrate that in standard supervised learning settings, IAM enhances generalization, achieving performance comparable to that of existing methods such as Sharpness-Aware Minimization. Furthermore, IAM exhibits efficacy in semi- and self-supervised learning scenarios, where the local inconsistency is computed from unlabeled data.

## 1. Introduction

Estimating the generalization gap and optimizing models to perform well on unseen data are central challenges in deep learning. Prior work has linked the flatness of the loss landscape to generalization and proposed sharpness-driven optimizers; however, sharpness—often instantiated as the largest eigenvalue of the loss Hessian—does not by itself reliably predict the generalization gap across settings (Keskar et al., 2017; Dinh et al., 2017; Li et al., 2018; Garipov et al., 2018; Foret et al., 2021; Kwon et al., 2021; Kim et al., 2022; Zhuang et al., 2022; Andriushchenko et al., 2023).

Alternatively, some studies examine output-based measures such as *disagreement* (Jiang et al., 2022) and *inconsistency* (Johnson & Zhang, 2023), which can correlate with the generalization gap under certain conditions. However, disagreement is non-differentiable and resists direct integration into training. Inconsistency is also impractical for a single model, since it requires aggregating outputs across multiple models and data splits at high computational cost.

In this work, we introduce *local inconsistency*, an information-geometric measure of output sensitivity in parameter space. Concretely, local inconsistency is defined as the worst-case (within an $\ell_2$ ball) KL divergence between the output distributions of a model and its perturbed counterpart. Crucially, it is (i) **computable from a single trained model** and (ii) **differentiable**, enabling both estimation and *direct regularization* within standard training pipelines. Furthermore, its computation (iii) **relies only on unlabeled data**, a key property that unlocks applications in label-constrained settings, including semi-/self-supervised learning.

We theoretically ground local inconsistency by connecting it to the Fisher information matrix (FIM) and the loss Hessian, showing that, under a local quadratic approximation, it is governed by the FIM's largest eigenvalue. This provides a complementary signal to traditional sharpness (e.g., $\lambda_{\max}(H)$), as we find that local inconsistency maintains a meaningful correlation with the generalization gap even in settings where sharpness measures falter.

Building on this, we propose *Inconsistency-Aware Minimization* (IAM)[1], which incorporates local inconsistency into the training objective. IAM inherits the practical advantages of single-model training while uniquely enabling *regularization from unlabeled data*. On CIFAR-10/100 supervised benchmarks, IAM matches or surpasses sharpness-aware

[1]Department of Computer Science, Hanyang University, Seoul, Korea [2]Agency for Defense Development, Daejeon, Korea. Correspondence to: Sungyoon Lee <sungyoonlee@hanyang.ac.kr>.

*Proceedings of the 43rd International Conference on Machine Learning*, Seoul, South Korea. PMLR 306, 2026. Copyright 2026 by the author(s).

[1]Code is available at https://github.com/heesung-k/IAM.

baselines. Crucially, its label-agnostic nature makes it a versatile regularizer for other learning paradigms; we show it boosts the performance of both the semi-supervised framework FixMatch and the self-supervised method SimCLR, demonstrating its broad applicability.

- **A computable and differentiable measure from unlabeled data.** We introduce *local inconsistency*, an information-geometric generalization measure that is *model-intrinsic* and *label-free*, making it practical both to estimate and to *regularize* during training.

- **Theory: links to FIM/Hessian and to prior inconsistency.** We formalize connections from *local inconsistency* to the FIM (and via Gauss–Newton to the Hessian) and discuss a relationship to Johnson & Zhang (2023), clarifying how local inconsistency complements inconsistency while avoiding the multi-model costs of prior inconsistency measures.

- **Method: IAM for labeled, semi-/self-supervised learning.** We develop IAM, which incorporates local inconsistency into the training objective. IAM achieves competitive or superior generalization to SAM in supervised tasks and, uniquely, *leverages unlabeled data* to improve semi- and self-supervised training.

## 2. Related Work

Understanding and improving generalization in deep neural networks, especially given their large capacity and tendency to overfit (Zhang et al., 2017), remains a central challenge. While networks can memorize random labels (Zhang et al., 2017) and learn simple patterns before noise (Arpit et al., 2017), phenomena like double descent (Nakkiran et al., 2021) and the inadequacy of uniform convergence theory (Nagarajan & Kolter, 2019) highlight the need for novel generalization measures beyond loss-based metrics.

Traditional measures like VC-dimension often fall short. While spectrally-normalized margin bounds (Bartlett et al., 2017) and PAC-Bayes approaches offer insights, no single measure consistently predicts generalization (Jiang* et al., 2020). Recently, disagreement (Jiang et al., 2022) and inconsistency (Johnson & Zhang, 2023) have shown promise, correlating well with the generalization gap, even when computed on unlabeled data. However, their reliance on training multiple models poses practical limitations for direct optimization in a single-model setup, underscoring the need for efficient, label-free, single-model generalization measures.

The geometry of the loss landscape, particularly the flatness of minima, has been extensively linked to generalization (Keskar et al., 2017; Li et al., 2018). However, the utility of sharpness as a sole predictor is debated due to issues like scale invariance (Dinh et al., 2017) and its correlation with training hyperparameters rather than true generalization (Andriushchenko et al., 2023). Indeed, some studies suggest that output inconsistency and instability can be more reliable predictors than sharpness (Johnson & Zhang, 2023). Information geometry has inspired a reparametrization-invariant sharpness measure (Jang et al., 2022), but these can be computationally expensive. This context motivates our exploration of "local inconsistency", an alternative geometric measure focusing on output sensitivity within a parameter neighborhood, computable from unlabeled data using a single model.

Various regularization techniques, both explicit (e.g., dropout (Srivastava et al., 2014), batch normalization (Santurkar et al., 2018), Mixup (Zhang et al., 2018)) and implicit (e.g., SGD's bias (Hardt et al., 2016; Soudry et al., 2018)), aim to improve generalization. Methods like Sharpness-Aware Minimization (SAM, (Foret et al., 2021)) and ASAM (Kwon et al., 2021) directly optimize for flat minima and have shown significant improvements. Despite their success, the precise role of sharpness in generalization remains an active area of research (Jiang* et al., 2020; Andriushchenko et al., 2023), further motivating the development of complementary approaches like our proposed IAM.

## 3. Background and Preliminaries

In this section, we briefly review fundamental concepts and notations essential for understanding our proposed metric and its theoretical connections. We focus on probabilistic classification models, information geometry, and aspects of the loss landscape.

### 3.1. Notation and Problem Setup

We consider probabilistic classification models. Let $x \in \mathcal{X}$ be a data point from the input space $\mathcal{X}$, and $y \in [C] = \{0, 1, \ldots, C-1\}$ be the corresponding class label, where $C$ is the total number of classes. The data pair $(x, y)$ is assumed to be drawn from an underlying distribution $\mathscr{D}$ over $\mathcal{X} \times [C]$. A model, parameterized by $\theta \in \mathbb{R}^m$, outputs a probability distribution over classes for a given input $x$. This is typically achieved by transforming a logit vector $z(x; \theta)$ through a softmax function: $f(x; \theta) = \mathrm{softmax}(z(x, \theta))$. Thus, $f(x; \theta) = [p(0|x; \theta), p(1|x; \theta), \ldots, p(C-1|x; \theta)]^\top$. Given a training dataset $Z_n = \{(x_i, y_i) : i = 1, \ldots, n\}$ drawn i.i.d. from $\mathscr{D}$, the model is typically trained by minimizing a loss function. For classification, the empirical Cross-Entropy (CE) loss will be written as follows:

$$L(\theta) = \frac{1}{n} \sum_{i=1}^{n} l_i(\theta),$$

where $l_i(\theta) = l(x_i, y_i; \theta) = -\log p(y_i|x_i; \theta)$.

## 3.2. Fisher Information Matrix and KL Divergence

The Fisher information matrix (FIM), $F(\theta)$, for the family of probability density $p(x, y; \theta) = p(x)p(y|x; \theta)$ parameterized by parameters $\theta$ is defined as

$$
\begin{aligned}
F(\theta) &= \mathbb{E}_x \left[ \mathbb{E}_{y \sim p(y|x;\theta)} \left[ \nabla_\theta l(x, y; \theta) \nabla_\theta l(x, y; \theta)^\top \right] \right] \\
&= \mathbb{E}_x \left[ \nabla_\theta z^\top \left( \text{diag}(f(x; \theta)) - f(x; \theta)f(x; \theta)^\top \right) \nabla_\theta z \right].
\end{aligned}
\tag{1}
$$

In practice, the expectation $\mathbb{E}_{p(x)}$ is often approximated by an empirical average over the available data (e.g., training data $\{x_i\}_{i=1}^n$ or unlabeled data).

The Kullback-Leibler (KL) divergence between the output distributions of a model with parameters $\theta$ and a slightly perturbed model $\theta + \delta$, $f(x; \theta)$ and $f(x; \theta + \delta)$, respectively, can be locally approximated using a second-order Taylor expansion with respect to $\delta$ as:

$$
\begin{aligned}
\mathbb{E}_x &\left[ \text{KL} \left( f(x; \theta) \| f(x; \theta + \delta) \right) \right] \\
&= \frac{1}{2} \delta^\top F(\theta) \delta + O(\|\delta\|^3).
\end{aligned}
\tag{2}
$$

## 3.3. Loss Hessian and Gauss-Newton Approximation

The geometry of the empirical loss surface $L(\theta)$ is described by its Hessian matrix $H(\theta) = \nabla_\theta^2 L(\theta)$. For the CE loss, the Hessian can be approximated by the Gauss-Newton (GN) matrix, $G(\theta)$. The second derivative of the per-sample CE loss $\ell_i(\theta)$ with respect to the logits $z_i = z(x_i; \theta)$, $\nabla_z^2 \ell_i(\theta) = \text{diag}(f(x_i; \theta)) - f(x_i; \theta)f(x_i; \theta)^\top$, depends only on the model's output probabilities $f(x_i; \theta)$. Consequently, the per-sample GN term, $G_i(\theta) = \nabla_\theta z_i^\top (\nabla_z^2 \ell_i) \nabla_\theta z_i$, is equivalent to the FIM contribution in Eq. (1). The empirical GN matrix, $G(\theta) = \frac{1}{n} \sum_{i=1}^n G_i(\theta)$, thus often termed the empirical FIM, provides a positive semi-definite approximation to $H(\theta)$:

$$
H(\theta) \approx G(\theta) = F(\theta)
$$

and is frequently used in optimization (Martens, 2020; Pascanu & Bengio, 2013).

# 4. Assessing Generalization Gap via Local Inconsistency

This section introduces our proposed measure, local inconsistency, designed to capture the generalization gap. We first define local inconsistency and elucidate its theoretical underpinnings by connecting it to the FIM and the loss Hessian. We then discuss its relationship with inconsistency (Johnson & Zhang, 2023). Finally, we present empirical results demonstrating the correlation between local inconsistency and the generalization gap, comparing it with other common measures.

## 4.1. Local Inconsistency, $S_\rho(\theta)$

We introduce local inconsistency, $S_\rho(\theta)$, defined as:

$$
S_\rho(\theta) = \max_{\|\delta\| \leq \rho} \mathbb{E}_{x \sim p(x)}[\text{KL}(f(x; \theta) \| f(x; \theta + \delta))],
\tag{3}
$$

which represents the sensitivity of the model's output distribution $f(x; \theta)$ with respect to the worst perturbations $\delta$, within a Euclidean ball of radius $\rho$ around the parameter $\theta$. Intuitively, a high value of $S_\rho(\theta)$ indicates that the model's output distribution is highly sensitive to small perturbations in parameter space. This sensitivity suggests potential instability or uncertainty in the model's predictions associated with the vicinity of $\theta$.

**Practical Advantages of $S_\rho$** Local inconsistency shares a practical advantage with sharpness-based measures (Keskar et al., 2017; Foret et al., 2021) in that it can be calculated using a **single** trained model. Furthermore, like disagreement (Jiang et al., 2022) and inconsistency (Johnson & Zhang, 2023), our metric can be estimated using only **unlabeled** data. A notable advantage over inconsistency and disagreement estimation is that evaluating $S_\rho$ does not require training multiple model instances derived from the same training procedure and is **directly regularizable**. This potentially makes $S_\rho$ more computationally efficient and practical to compute, especially when model training is resource-intensive.

## 4.2. Connection to FIM and Hessian

The relationship between our metric $S_\rho$ and the Fisher Information Matrix (FIM) can be established by leveraging the local quadratic approximation of the KL divergence, as outlined in Section 3. With this quadratic approximation, we can approximate $S_\rho(\theta)$ with the maximum eigenvalue of FIM, scaled by $\rho^2/2$:

$$
\begin{aligned}
S_\rho(\theta) &\approx \max_{\|\delta\| \leq \rho} \frac{1}{2} \delta^\top F(\theta) \delta \\
&= \frac{1}{2} (\rho v_{\max})^\top F(\theta) (\rho v_{\max}) \\
&= \frac{1}{2} \rho^2 \lambda_{\max}(F(\theta)),
\end{aligned}
\tag{4}
$$

where $v_{\max}$ is the eigenvector corresponding to the largest eigenvalue $\lambda_{\max}$ of $F(\theta)$. Remarkably, this approximation requires only the model $\theta$ and unlabeled data (used to compute the expectation).

The Fisher Information Matrix $F(\theta)$, to which $S_\rho(\theta)$ is related via its maximum eigenvalue, also connects to the Hessian of the loss function $H(\theta)$. As detailed in Section 3, for Negative Log Likelihood losses such as CE, the Hessian can be approximated by the Gauss-Newton matrix $G(\theta)$, equivalent to empirical FIM computed using training data.

Consequently, when calculating $S_\rho(\theta)$ using the training data, it approximates $\frac{1}{2}\rho^2\lambda_{\max}(G(\theta))$. Given that $G(\theta)$ often provides a good approximation to the true loss Hessian near a local minimum, $S_\rho(\theta)$ therefore offers insights into the maximum curvature of the loss landscape in that vicinity.

### 4.3. Local Inconsistency and Generalization Bounds

Under near interpolation, which is a standard regime in modern deep learning (Zhang et al., 2017), the empirical Hessian splits into a Fisher/Gauss–Newton term plus a small residual, which lets us replace $\lambda_{\max}(H_S(\theta))$ with $\lambda_{\max}(F_S(\theta))$ up to a spectral slack.

**Theorem 4.1** (FIM-based generalization bound). *Under the same assumptions of Theorem 3.1 of Luo et al. (2024), for any $\xi \in (0,1)$ and $\rho > 0$, with probability at least $1 - \xi$ over the choice of $S \sim \mathscr{D}$, we have*

$$L_{\mathscr{D}}(\theta) \leq L_S(\theta) + \frac{\rho^2}{2}\Big(\lambda_{\max}\big(F_S(\theta)\big) + \varepsilon_R\Big) + \frac{Cm^{3/2}\rho^3}{6}$$
$$+ O\left(\frac{1}{\sqrt{n}}\right),$$

*where $n$ is the number of samples.*

In particular, at (near) interpolation ($\varepsilon_R \approx 0$), the residual vanishes and the curvature term reduces to $\lambda_{\max}(F_S(\theta))$ with no degradation. We defer the exact complexity term and proof to Appendix A.

This bound suggests that minimizing a combination of $L_S(\theta)$ and the local inconsistency $S_\rho(\theta)$ lowers the upper bound on the true risk $L_{\mathscr{D}}(\theta)$. This provides a theoretical motivation for our Inconsistency-Aware Minimization (IAM) framework, which aims to find solutions that are accurate on the training data while keeping output sensitivity low in parameter space, as measured by $S_\rho(\theta)$.

### 4.4. Relation with Inconsistency in Johnson & Zhang (2023)

Local inconsistency exhibits an interesting relationship to the inconsistency in Johnson & Zhang (2023) defined as:

$$\mathcal{C}_P = \mathbb{E}_{Z_n}\mathbb{E}_{\theta,\theta'\sim\Theta_{P|Z_n}}\mathbb{E}_{x\sim p(x)}[\mathrm{KL}(f(x;\theta)\|f(x;\theta'))].$$

We consider the conditional inconsistency for a fixed $Z_n$, denoted $\mathcal{C}_{P|Z_n}$, without outer expectation. Then our proposed metric, $S_\rho(\theta_{Z_n})$, is approximately proportional to the conditional inconsistency $\mathcal{C}_{P|Z_n}$:

$$\frac{m}{2C}\mathcal{C}_{P|Z_n} \lesssim S_\rho(\theta_{Z_n}) \lesssim \frac{m}{2}\mathcal{C}_{P|Z_n}, \qquad (5)$$

under certain assumptions, such as assuming the parameter posterior $\Theta_{P|Z_n}$ as a distribution with isotropic covariance and $\theta_{Z_n}$ as mean. This connection arises because both metrics are related to the local geometry captured by the FIM

at $\theta_{Z_n}$, with $S_\rho$ being linked to its maximum eigenvalue and $\mathcal{C}_{P|Z_n}$ to its trace. Practically, the eigenspectra of the FIM of a neural network are observed to be dominated by a few large eigenvalues (specifically related to the number of classes, $C$ in classification task) while remaining eigenvalues are near zero (Sagun et al., 2017; Papyan, 2018; 2019; 2020; Karakida et al., 2019; 2021). This observation indicates that the ratio $\lambda_{\max}(F(\theta))/\mathrm{Tr}(F(\theta))$ is larger than $\frac{1}{C}$ ($C \ll m$). For detailed derivation, please see Appendix B.

### 4.5. Estimating Local Inconsistency $S_\rho(\theta)$

Directly computing $S_\rho(\theta)$ requires solving the maximization problem over the high-dimensional parameter perturbation $\delta$. For deep neural networks, finding the exact maximum within the $L_2$-ball of radius $\rho$ is generally intractable. Therefore, we employ numerical approximation methods.

For small perturbations $\delta$, the expected KL divergence can be accurately approximated by a second-order Taylor expansion involving the Fisher Information Matrix (FIM), $F(\theta)$, as Eq. (2) of Section 3. Under quadratic approximation, as discussed in Section 4.2, the optimal perturbation $\delta^* = \rho v_{\max}$, the maximum value is then $S_\rho(\theta) = \frac{1}{2}\rho^2\lambda_{\max}$, and the gradient of the approximated KL divergence with respect to $\delta$ is $F(\theta)\delta$.

This connection motivates, instead of the usual Projected Gradient Ascent update $\delta \leftarrow \Pi_{\{\delta:\|\delta\|\leq\rho\}}(\delta + \eta F(\theta)\delta)$, an iterative gradient ascent approach that updates:

$$\delta_{k+1} = \frac{\rho}{\|F(\theta)\delta_k\|}F(\theta)\delta_k, \quad \delta_0 = \varepsilon \sim \mathcal{N}\left(0, \frac{\sigma^2}{m}I_m\right),$$

where $\sigma^2$ is initial noise scale. Iterative gradient ascent is precisely one iteration of the Power Iteration method used to find the dominant eigenvector of $F(\theta)$.

**Algorithm for estimating $S_\rho(\theta)$** Based on the above, we propose Algorithm 1 to estimate $S_\rho(\theta)$. This algorithm performs $K$ steps of normalized gradient ascent (effectively, Power Iteration under the quadratic approximation) to find an approximate maximizing perturbation $\delta^*$. Algorithm 1 requires $K$ gradient computations.

---

**Algorithm 1** Estimation of $S_\rho(\theta)$

---

**Input:** model parameter $\theta \in \mathbb{R}^m$, noise scale $\sigma^2$, radius $\rho > 0$, number of steps $K \geq 1$
**Initialize** $\delta_0$ randomly with $\mathcal{N}(0, \frac{\sigma^2}{m}I_m)$
**for** $k = 0$ to $K - 1$ **do**
    Compute $g_k = \nabla_\delta\mathbb{E}_x\mathrm{KL}(f(x;\theta)\|f(x;\theta+\delta))|_{\delta=\delta_k}$
    Update perturbation: $\delta_{k+1} = \rho\frac{g_k}{\|g_k\|}$
**end for**
**return** $\mathbb{E}_{x\sim p(x)}\mathrm{KL}(f(x;\theta)\|f(x;\theta+\delta_K))$

---

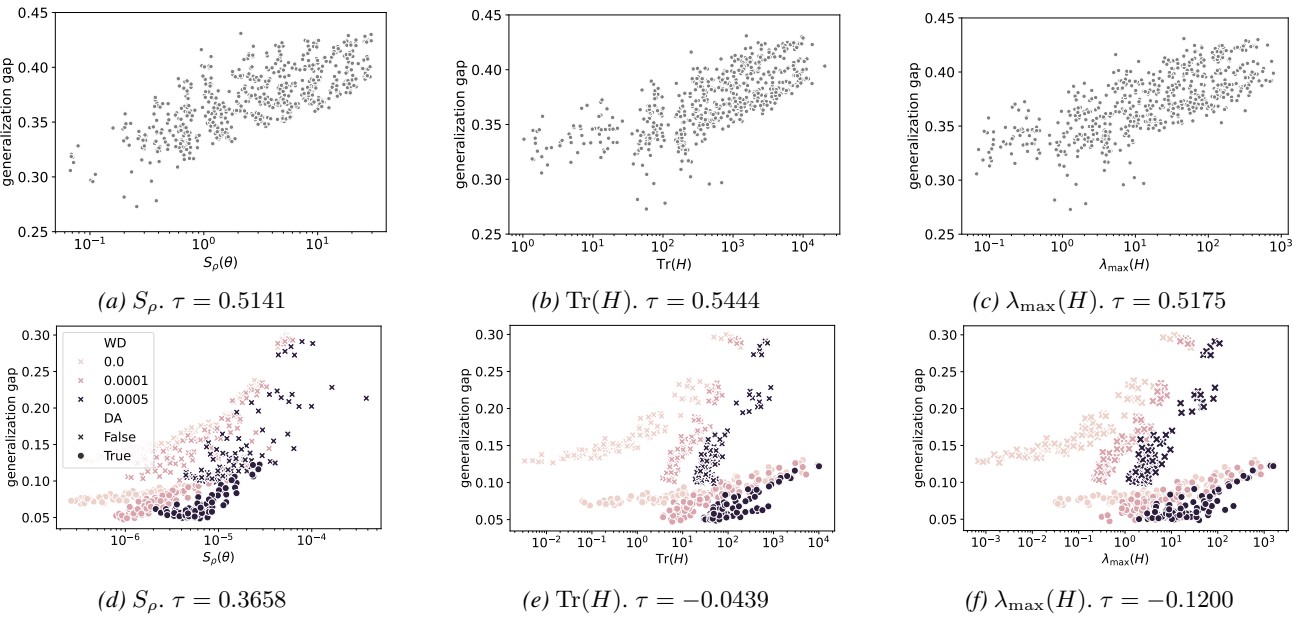

*Figure 1.* Local inconsistency and sharpness measures vs the generalization gap.

## 4.6. Empirical Results

To compare the predictive capability of $S_\rho$ for the generalization gap with traditional loss-based sharpness measures, we conducted experiments on CIFAR-10. We trained two distinct architectures, a 6-layer CNN (6CNN) and a Wide Residual Network (WRN28-2) (Zagoruyko & Komodakis, 2016), under various hyperparameter settings (details in Appendix F). $S_\rho$ was estimated using a disjoint, unlabeled dataset. For comparison, we also computed two common sharpness measures: the trace, $\mathrm{Tr}(H)$, and the maximum eigenvalue, $\lambda_{\max}(H)$.

Figure 1 presents scatter plots of these metrics against the generalization gap (test error - train error), with Kendall's Tau ($\tau$) reported for each. For the simpler 6CNN model (top row), $S_\rho$ ($\tau = 0.5141$) exhibited a positive correlation with the generalization gap, comparable to $\mathrm{Tr}(H)$ ($\tau = 0.5444$) and $\lambda_{\max}(H)$ ($\tau = 0.5175$). This suggests that for smaller models, various geometric measures may similarly capture aspects of generalization.

However, for the larger WRN28-2 model under varying data augmentation and weight decay settings (bottom row), a more nuanced behavior emerged. As noted by Andriushchenko et al. (2023), different training configurations—specifically combinations of weight decay and data augmentation—can form distinct solution subgroups. $\mathrm{Tr}(H)$ and $\lambda_{\max}(H)$ show positive correlations only within such subgroups but exhibit negative overall correlations globally ($\tau = -0.0439$ and $\tau = -0.1200$, respectively). In stark contrast, our $S_\rho$ maintained a positive, albeit reduced, global correlation ($\tau = 0.3658$).

This divergence, particularly with larger models and data augmentation, suggests that the output-based formulation of local inconsistency is more robust to the absolute scale effects that vary across models and data-augmentation settings than traditional Hessian-based sharpness. While the predictive utility of sharpness metrics can be confounded by these subgroup effects, $S_\rho$ demonstrates more consistent global predictiveness, hinting at its potential as a more robust generalization indicator in complex training scenarios.

## 5. Inconsistency-Aware Minimization (IAM): Incorporating Local Inconsistency into the Objective

Our empirical findings suggest that local inconsistency, $S_\rho(\theta)$ defined in Eq. (3), correlates with the generalization gap. This motivates its use as a regularizer to guide the optimization towards solutions that not only fit the training data, but also exhibit low sensitivity in their output distributions with respect to parameter perturbations. We propose two strategies to incorporate local inconsistency into the training objective.

1. **Direct Regularization (IAM-D)**: This approach directly penalizes local inconsistency by adding it to the standard training loss $L(\theta)$:

$$L_{\text{IAM-D}}(\theta) = L(\theta) + \beta S_\rho(\theta) \tag{6}$$

where $\beta > 0$ is a hyperparameter balancing the trade-off. This objective seeks parameter values $\theta$ for which the model outputs are consistent across the neighborhood.

**Algorithm 2** Inconsistency-Aware Minimization

---

**Input:** Initial model parameters $\theta^0$; Learning rate $\eta$; Base optimizer $\mathcal{O}$; Neighborhood size $\rho$; training set $Z_n$; Batch size $b$; Number of steps $K$ for Algorithm 1.
**while** not converged **do**
    Sample batch $\{(x_i, y_i)\}_{i=1}^b$.
    Compute $\delta_K$ from Algorithm 1 using current $\theta$, $\rho$, and data $\{x_i\}_{i=1}^b$.
    **if** IAM-S **then**
        Compute gradient $g = \nabla_\theta L(\theta)|_{\theta+\delta_K}$
    **else if** IAM-D **then**
        Compute gradient $g = \nabla_\theta(L(\theta) + \beta S_\rho(\theta))$
    **end if**
    $\theta \leftarrow \mathcal{O}(\theta, g, \eta)$
**end while**
**Return** optimized parameters $\theta$.

---

## 2. SAM-like Approach (IAM-S):

**2. SAM-like Approach (IAM-S)**: Inspired by SAM (Foret et al., 2021), this method aims to find parameters $\theta$ that reside in a neighborhood of uniformly low loss by minimizing the loss at an adversarially perturbed point $\theta + \delta^*$:

$$L_{\text{IAM-S}}(\theta) = L(\theta + \delta^*), \tag{7}$$

$\delta^* = \arg\max_{\|\delta\| \le \rho} \frac{1}{n} \sum_{i=1}^n \text{KL}(f(x_i, \theta) \| f(x_i, \theta + \delta))$ is the perturbation that maximizes the local inconsistency term with current mini-batch. Note that the objective minimizes the original loss $L$ at the perturbed point $\theta + \delta$:

$$L(\theta + \delta) \approx L(\theta) + \delta^\top \nabla_\theta L(\theta) + \frac{1}{2}\delta^\top G(\theta)\delta.$$

Note that our perturbation $\delta$ and $-\delta$ have equal probability of being sampled. Consequently, the symmetry mitigates the influence of the first-order term $\delta^\top \nabla_\theta L(\theta)$ in expectation, suggesting IAM-S implicitly minimizes the principal eigenvalues of $G(\theta)$ (equivalent to empirical FIM).

### 5.1. Algorithm for Inconsistency-Aware Minimization

Optimizing $L_{\text{IAM-D}}(\theta)$ and $L_{\text{IAM-S}}(\theta)$ involves a min-max procedure. The inner maximization to find $\delta^*$ (i.e., computing $S_\rho(\theta)$ and the corresponding $\delta^*$) is performed using an Algorithm 1 with current mini-batch, typically for $K = 1$ to match the number of additional gradient computations to that of SAM. For **IAM-D**, we simply add the regularization term $\beta S_\rho(\theta)$ to the original loss $L(\theta)$ and compute the gradient $g = \nabla_\theta(L(\theta) + \beta S_\rho(\theta))$. In the case of **IAM-S**, the outer minimization step calculates the gradient of the loss at the perturbed point $\theta + \delta_K$. Following the approximation used in SAM, we drop the second-order terms arising from the derivative $\nabla_\theta \delta$ (see Appendix D for theoretical justification for IAM-D). Finally, a base optimizer $\mathcal{O}$ (e.g., SGD or Adam) updates $\theta$ using the computed gradient $g$, as summarized in Algorithm 2.

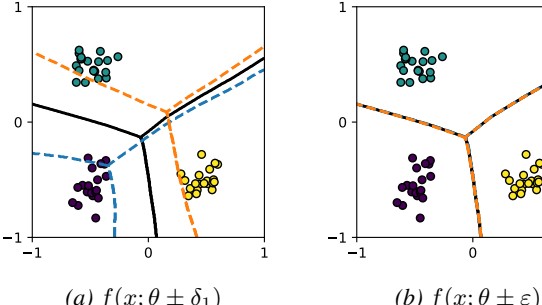

*(a) $f(x; \theta \pm \delta_1)$*        *(b) $f(x; \theta \pm \varepsilon)$*

*Figure 2.* A synthetic classification example. The black, blue, orange lines correspond to decision boundaries of the NN with trained parameter, and parameter perturbed by $\pm\delta_1$ (a) or $\pm\varepsilon$ (b).

Figure 2 intuitively shows the role of perturbation $\delta_1$ with decision boundaries of a Neural Network (NN) for two-dimensional synthetic data generated from mixtures of three Gaussians. As Jang et al. (2022) show, the density change occurs mainly along the principal eigenvectors of FIM; thus, $\delta_1$ is aligned with principal eigenvectors of FIM and causes a meaningful output distribution shift while the noise perturbation $\varepsilon$ with the same norm does not. See Appendix C for a detailed analysis of the effectiveness of $\delta_1$ in this setting. Moreover, as $K$ increases, IAM benefits from a more accurate estimation of local inconsistency, offering a trade-off between performance and cost (see Appendix E.1).

### 5.2. Empirical Evaluation in Supervised Learning

We evaluate the performance of IAM against SGD, SAM, and ASAM (Kwon et al., 2021) on standard image classification tasks. Wide ResNet (WRN) (Zagoruyko & Komodakis, 2016) serves as the baseline model, trained on CIFAR-{10, 100}, F-MNIST, and SVHN with standard data augmentations. Specifically, we employ WRN-16-8 for CIFAR-{10, 100} and WRN-28-10 for F-MNIST and SVHN. Optimal hyperparameters were determined via a grid search: for IAM-D, we set $\beta = 1.0, \rho = 0.1$ on CIFAR-10 and $\beta = 10.0, \rho = 0.1$ on CIFAR-100; for IAM-S, we used $\rho = 0.1$ and $\rho = 0.5$ for CIFAR-10 and CIFAR-100, respectively. To ensure a fair comparison under a fixed computational budget, we report the best test error for SGD from either 200 or 400 epochs. Detailed experimental settings are listed in Appendix F. For additional results of ViT (Dosovitskiy et al., 2021) on CIFAR-10 and ImageNet, please refer to Table 8 and Table 6 in Appendix E.3.

Table 1 summarizes the test error rates. Both IAM-D and IAM-S not only reduce test error compared to SGD but also achieve performance comparable to SAM and ASAM on overall datasets. Notably, on the complex CIFAR-100 dataset, IAM-S outperforms SAM by a margin of 0.81%, demonstrating its effectiveness in challenging scenarios.

*Table 1.* Test Error (mean $\pm$ stderr) of SGD, SAM, ASAM, and IAM across datasets.

| Dataset | SGD | SAM | ASAM | IAM-D | IAM-S |
|---|---|---|---|---|---|
| CIFAR-10 | 3.68 $\pm 0.04$ | 3.31 $\pm 0.01$ | **3.15** $\pm 0.02$ | 3.28 $\pm 0.06$ | 3.28 $\pm 0.03$ |
| CIFAR-100 | 19.17 $\pm 0.19$ | 17.63 $\pm 0.12$ | 17.15 $\pm 0.11$ | 17.16 $\pm 0.03$ | **16.82** $\pm 0.01$ |
| F-MNIST | 4.45 $\pm 0.05$ | 4.13 $\pm 0.02$ | 4.11 $\pm <0.01$ | 4.13 $\pm 0.04$ | **4.10** $\pm 0.05$ |
| SVHN | 3.82 $\pm 0.06$ | 3.47 $\pm 0.09$ | 3.24 $\pm 0.04$ | **3.13** $\pm 0.06$ | **3.13** $\pm 0.01$ |

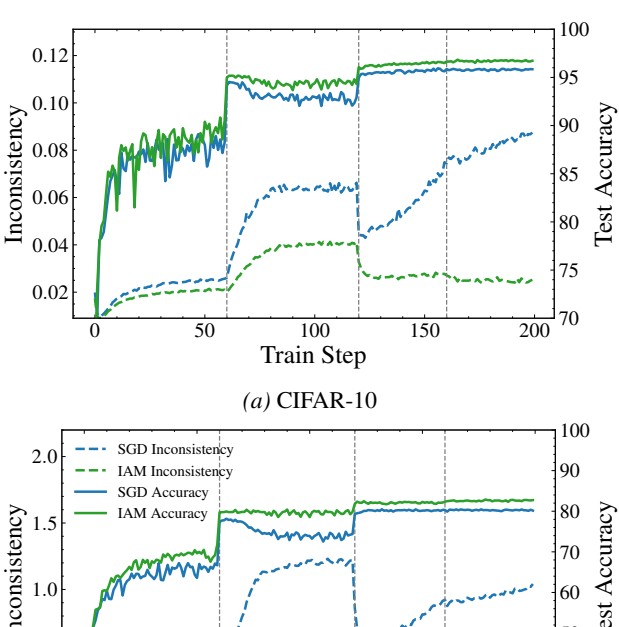

*(a)* CIFAR-10

*(b)* CIFAR-100

*Figure 3.* The evolution of the local inconsistency $S_\rho(\theta)$ and test accuracy with SGD and IAM-D.

We further analyze the training dynamics in Figure 3, which illustrates the evolution of local inconsistency $S_\rho(\theta)$ and test accuracy for SGD and IAM-D. IAM-D effectively suppresses the increase in $S_\rho(\theta)$ and mitigates overfitting. This effect is particularly evident after the learning rate decay points, where the test accuracy of SGD tends to degrade while its inconsistency rebounds. In contrast, IAM-D maintains a stable $S_\rho(\theta)$ below that of SGD throughout the training. These observations suggest that minimizing local inconsistency helps confine the model to parameter regions with smoother output distributions, which correlates with the generalization improvements observed in Table 1.

To verify the scalability of our approach, we extend our evaluation to the large-scale ImageNet dataset using a ResNet-50 architecture. Following the experimental setting of Foret et al. (2021), we train the model with a batch size of 1024,

*Table 2.* Top-1 and Top-5 error (mean $\pm$ stderr) of ResNet-50 trained 200 epochs on ImageNet.

| | Top-1 | Top-5 |
|---|---|---|
| SGD | 22.66 $\pm 0.12$ | 6.51 $\pm 0.06$ |
| SAM | 21.80 $\pm 0.12$ | 5.99 $\pm 0.04$ |
| IAM-D | **21.36** $\pm 0.06$ | **5.70** $\pm 0.02$ |
| IAM-S | 21.72 $\pm 0.07$ | 5.90 $\pm 0.02$ |

initial learning rate of 0.2, and basic augmentations. We set $\rho = 0.2$ for IAM-S, $\rho = 0.1$ for IAM-D, and $\rho = 0.05$ for SAM. We report the best score achieved by SGD at either 200 epochs or 400 epochs. As shown in Table 2, IAM-D and IAM-S outperform the standard SGD baseline in both Top-1 and Top-5 error. In particular, IAM-D outperforms the stronger SAM baseline. See Table 5 in Appendix E.2 for the results of each epochs.

While SAM relies on the gradient of the training loss, IAM derives perturbations via the KL divergence. The generalization performance of IAM-S empirically demonstrates that a single-step perturbation $\delta_1 \approx F(\theta)\varepsilon$ is not merely stochastic; rather, it effectively captures the principal eigenspace of the FIM with minimal computational effort ($K = 1$). This observation aligns with Explicit Jacobian Regularization (Lee et al., 2023), which showed that random noise becomes a meaningful perturbation when projected onto the Jacobian's column space. The same mechanism applies here: multiplying $\varepsilon$ by $F(\theta)$ weights each eigendirection by its eigenvalue, so a single step amplifies the dominant curvature directions while suppressing the rest.

### 5.3. IAM for Learning with Limited or No Explicit Labels

A key advantage of local inconsistency is its computability from unlabeled data, making IAM well-suited for scenarios with limited or no explicit supervision. We demonstrate this in semi-supervised and self-supervised learning settings. IAM-D can be seamlessly "plugged in" to complex pipelines like FixMatch (Sohn et al., 2020) or SimCLR (Chen et al., 2020) by adding penalty term $\beta S_\rho(\theta)$ to the original objective. Detailed experimental settings are listed in Appendix F.

*Table 3.* Test error (mean ± stderr) with semi-supervised setting on CIFAR-10 and CIFAR-100

| | CIFAR-10 | | CIFAR-100 | |
|---|---|---|---|---|
| | 250 labels | 4000 labels | 2500 labels | 10000 labels |
| SGD | 63.82 ± 0.18 | 22.45 ± 0.40 | 68.91 ± 0.43 | 45.94 ± 0.35 |
| SAM | 63.91 ± 0.18 | 19.95 ± 0.22 | 69.53 ± 0.79 | 43.30 ± 0.11 |
| IAM-D | **61.77** ± 0.09 | **15.07** ± 0.14 | **66.98** ± 0.01 | **40.02** ± 0.13 |
| FixMatch | 6.26 ± 0.39 | 4.10 ± 0.17 | 32.84 ± 0.40 | 22.93 ± 0.05 |
| FixMatch + IAM-D | **5.30** ± 0.08 | **3.88** ± 0.02 | **28.95** ± 0.59 | **21.99** ± 0.04 |

**Semi-Supervised Learning.** We demonstrate the advantage of IAM in a label-scarce setting on CIFAR-{10, 100}. Our method, IAM-D, optimizes a joint objective: the standard cross-entropy loss on the labeled subset, plus the local inconsistency penalty computed over the entire mini-batch (both labeled and unlabeled samples). The results in Table 3 show that IAM-D consistently outperforms both SGD and SAM. Furthermore, to highlight its versatility, we integrated IAM-D into the strong FixMatch framework (Sohn et al., 2020). This combination significantly lowers the test error both on CIFAR-10 and CIFAR-100, demonstrating that IAM-D can serve as an effective plug-and-play regularizer to enhance state-of-the-art semi-supervised learning methods.

This approach contrasts with methods like SAM, which can only promote flatness over the small, labeled subset. Simply applying SAM to labeled loss of FixMatch fails to improve generalization (see Appendix E.4). A critical insight is that flatness measured on a sparse set of labeled points may not reflect true flatness across the entire data distribution. By leveraging second-order information from abundant unlabeled data, IAM-D seeks a more generalizable minimum.

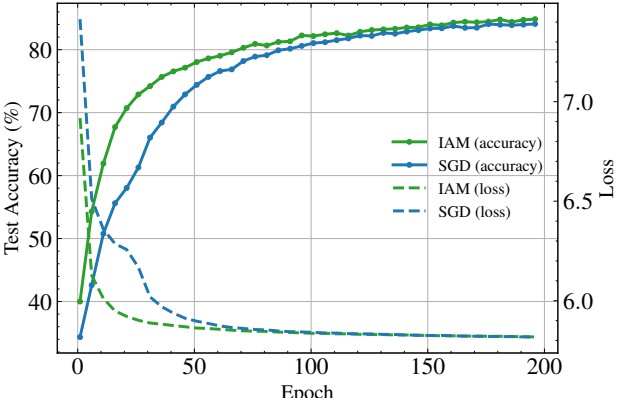

*Figure 4.* Test accuracy on linear probe and SimCLR training loss for ResNet-18 on CIFAR-10, comparing SimCLR trained with SGD (SimCLR-SGD) versus SimCLR with IAM-D (SimCLR-IAM).

**Self-Supervised Learning (SSL).** The label-agnostic nature of IAM makes it directly applicable to SSL objectives. We integrated IAM-D into the SimCLR framework (Chen et al., 2020), training a ResNet-18 (He et al., 2016) encoder on CIFAR-10. Performance was evaluated using linear probing. The local inconsistency term for IAM-D was computed using the model's projection-head outputs. Figure 4 shows that SimCLR trained with IAM-D (SimCLR-IAM) achieves higher test accuracy on the downstream linear classification task compared to vanilla SimCLR (SimCLR-SGD). Furthermore, SimCLR-IAM tends to converge faster in terms of test error and also minimizes the SimCLR training loss more rapidly, despite the additional local inconsistency regularization. This suggests that controlling local inconsistency is beneficial even when no explicit labels are available during representation learning.

## 6. Conclusion

In this work, we introduced "local inconsistency," a novel information-geometric generalization measure computable from a single model using only unlabeled data. We theoretically linked it to the Fisher Information Matrix (FIM) and the loss Hessian. Empirically, local inconsistency correlates with the generalization gap and exhibits distinct characteristics from traditional sharpness-based metrics.

Based on this, we proposed Inconsistency-Aware Minimization (IAM), an optimization framework that directly incorporates local inconsistency into the training objective. IAM enhances generalization in supervised learning, matching or exceeding that of Sharpness-Aware Minimization (SAM). Crucially, IAM proves effective in semi- and self-supervised learning by leveraging unlabeled data for local inconsistency computation, improving performance in label-scarce settings.

These findings offer a practical and theoretically-grounded approach to improving model generalization, particularly valuable in real-world applications where labeled data is limited. Future research could focus on exploring the scalability and applicability of IAM to a wider array of modern model architectures and other tasks or on developing a computationally efficient version of IAM.

## Impact Statement

This paper presents work whose goal is to advance the field of Machine Learning. There are many potential societal consequences of our work, none which we feel must be specifically highlighted here.

## Acknowledgments

We thank the anonymous reviewers for insightful reviews. This work was partially supported by Institute of Information & communications Technology Planning & Evaluation (IITP) grants (RS-2020-II201373, Artificial Intelligence Graduate School Program (Hanyang University); RS-2023-002206284, Artificial intelligence for prediction of structure-based protein interaction reflecting physicochemical principles); the BK21 FOUR (Fostering Outstanding Universities for Research) project; NRF2024S1A5C3A02043653, Socio-Technological Solutions for Bridging the AI Divide: A Blockchain and Federated Learning-Based AI Training Data Platform) and Korea Institute for Advanced Study (KIAS) grant funded by the Korean government (MSIT).

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

## A. Proof of the FIM-based generalization bound

We provide a self-contained derivation of the FIM-form bound stated in Theorem 4.1. Throughout, let $L_S(\theta) = \frac{1}{n}\sum_{i=1}^n \ell(f(x_i;\theta), y_i)$ be the empirical cross-entropy with logits $z(x;\theta) \in \mathbb{R}^C$, probabilities $f(x;\theta) = \mathrm{softmax}(z)$, and $J(x;\theta) := \nabla_\theta z(x;\theta) \in \mathbb{R}^{C \times m}$. $\|\cdot\|$ denotes the Euclidean norm for vectors and the spectral norm for matrices. We write $H_S(\theta) := \nabla^2 L_S(\theta)$ and define the empirical Fisher

$$F_S(\theta) := \frac{1}{n}\sum_{i=1}^n J_i^\top\big(\mathrm{diag}(f(x_i;\theta)) - f(x_i;\theta)f(x_i;\theta)^\top\big)J_i,$$

where $J_i := J(x_i;\theta)$.

**Assumption (near interpolation).** There exists $\varepsilon_R \geq 0$ such that the residual

$$R_S(\theta) := \frac{1}{n}\sum_{i=1}^n \sum_{k=1}^C (f(x_i;\theta) - y_i)_k \, \nabla_\theta^2 z_k(x_i;\theta) \quad \text{satisfies} \quad \|R_S(\theta)\| \leq \varepsilon_R. \tag{A1}$$

### Step 1: Hessian–FIM decomposition for softmax–CE

**Lemma A.1** (Gauss–Newton (=FIM) + residual). *For each sample $i$, with loss $\ell_i := \ell(f(x_i;\theta), y_i)$,*

$$\nabla_\theta \ell_i = J_i^\top(f(x_i;\theta) - y_i)$$

$$\nabla_\theta^2 \ell_i = J_i^\top\big(\mathrm{diag}(f(x_i;\theta)) - f(x_i;\theta)f(x_i;\theta)^\top\big)J_i + \sum_{k=1}^C (f(x_i;\theta) - y_i)_k \, \nabla_\theta^2 z_k(x_i;\theta).$$

*Averaging over $i$ yields $H_S(\theta) = F_S(\theta) + R_S(\theta)$.*

*Proof.* Since $\ell(p, y) = -\sum_k y_k \log p_k$ and $p = \mathrm{softmax}(z)$, $\frac{\partial \ell}{\partial z} = p - y$. By the chain rule, $\nabla_\theta \ell_i = J_i^\top(f(x_i;\theta) - y_i)$. Differentiating once more,

$$\nabla_\theta^2 \ell_i = J_i^\top\Big(\frac{\partial f(x_i;\theta)}{\partial z_i}\Big)J_i + \sum_{k=1}^C \Big(\frac{\partial \ell_i}{\partial z_{ik}}\Big)\nabla_\theta^2 z_k(x_i, \theta),$$

and $\frac{\partial f(x_i;\theta)}{\partial z_i} = \mathrm{diag}(f(x_i;\theta)) - f(x_i;\theta)f(x_i;\theta)^\top$ for softmax. Using $\frac{\partial \ell_i}{\partial z_{ik}} = (f(x_i;\theta) - y_i)_k$ gives the stated identity. Averaging over $i$ completes the proof. □

### Step 2: Spectral control via Weyl's inequality

**Lemma A.2** (Hessian vs. FIM eigenvalues). *If $H_S = F_S + R_S$ with $F_S, R_S$ symmetric, then*

$$\lambda_{\max}(H_S) \leq \lambda_{\max}(F_S) + \|R_S\|.$$

Combining Lemma A.1 with Assumption (A1) and Lemma A.2 gives

$$\lambda_{\max}\big(H_S(\theta)\big) \leq \lambda_{\max}\big(F_S(\theta)\big) + \varepsilon_R. \tag{8}$$

### Step 3: From the Hessian-based bound to the FIM form

We recall the Hessian-based bound of Luo et al. (2024) (Theorem 3.1) under the assumption that the loss function is bounded by $L$, the third-order partial derivative of the loss function is bounded by $C$, and $L_{\mathscr{D}}(\theta) \leq \mathbb{E}_{\varepsilon \sim \mathcal{N}(0,\sigma^2 I_m)} L_{\mathscr{D}}(\theta + \varepsilon)$.

$$L_{\mathscr{D}}(\theta) \leq L_S(\theta) + \frac{m\sigma^2}{2}\lambda_{\max}\big(H_S(\theta)\big) + \frac{Cm^3\sigma^3}{6} \tag{9}$$

$$+ \frac{L}{2\sqrt{n}}\sqrt{m\log\big(1 + \frac{\|\theta\|^2}{\rho^2}\big) + 2\log\frac{1}{\xi} + 4\log(n+m) + O(1)}. \tag{10}$$

**Theorem A.3** (FIM-based generalization bound; Theorem 4.1). *Assume that the loss function is bounded by L, the third-order partial derivative of the loss function is bounded by C, and $L_{\mathscr{D}}(\theta) \leq \mathbb{E}_{\varepsilon \sim \mathcal{N}(0,\sigma^2 I_m)} L_{\mathscr{D}}(\theta + \varepsilon)$. For any $\xi \in (0,1)$ and $\rho > 0$, with probability at least $1 - \xi$ over the choice of $S \sim \mathscr{D}$, we have*

$$L_{\mathscr{D}}(\theta) \leq L_S(\theta) + \frac{\rho^2}{2}\left(\lambda_{\max}\big(F_S(\theta)\big) + \varepsilon_R\right) + \frac{Cm^{3/2}\rho^3}{6}$$
$$+ \frac{L}{2\sqrt{n}}\sqrt{m\log\big(1 + \frac{\|\theta\|^2}{\rho^2}\big) + 2\log\frac{1}{\xi} + 4\log(n+m) + O(1)}, \tag{11}$$

*where n is the number of samples and $\rho = \sqrt{m}\sigma$.*

In particular, at (near) interpolation where $\varepsilon_R \approx 0$ ($L_S(\theta) \approx 0$), the residual vanishes and the curvature term reduces to $\lambda_{\max}(F_S(\theta))$ without degradation.

*Proof.* Substitute (8) into (9). □

## B. Relation between Local Inconsistency and Inconsistency

This section outlines an approximate derivation relating the model output inconsistency $\mathcal{C}_P$, as defined by Johnson & Zhang (2023), to the local sensitivity metric $S_\rho(\theta)$ defined previously. we first show simple demonstrations that these two metrics are related primarily through the Fisher Information Matrix (FIM), under specific assumptions like isotropic covariance. We then show results with anisotropic covariance.

### Definitions

- **Inconsistency** ($\mathcal{C}_P$): Measures the average difference (in terms of KL divergence) between the outputs of models generated by a stochastic training procedure $P$ applied to the same training data $Z_n$. The average is taken over draws of the training data $Z_n$ and pairs of models $(\Theta, \Theta')$ drawn from the conditional distribution $\Theta_{P|Z_n}$.

$$\mathcal{C}_P = \mathbb{E}_{Z_n}\mathbb{E}_{\theta,\theta' \sim \Theta_{P|Z_n}}\mathbb{E}_x[\mathrm{KL}(f(x;\theta)\|f(x;\theta'))].$$

  Here, $\Theta_{P|Z_n}$ denotes the distribution over parameters resulting from applying procedure $P$ to dataset $Z_n$.

- **Local Inconsistency** ($S_\rho(\theta)$): Measures the expected maximum change in the model's output distribution within a $\rho$-radius ball around a specific parameter vector $w$. For consistency with the derivation below, we use the form where the expectation is inside the maximization.

$$S_\rho(\theta) = \max_{\|\delta\|\leq\rho} \mathbb{E}_x[\mathrm{KL}(f(x;\theta)\|f(x;\theta+\delta))].$$

  Here, $\delta \in \mathbb{R}^m$ is a perturbation to the parameters $w$.

**Assumptions** The following derivation relies on several key assumptions:

1. **Isotropic Covariance Posterior Assumption**: For a given training set $Z_n$, the conditional parameter distribution $\Theta_{P|Z_n}$ can be approximated by an isotropic distribution centered at a specific parameter vector $\theta_{Z_n}$ derived from $Z_n$: $\mathbb{E}[\Theta_{P|Z_n}] = \theta_{Z_n}, \mathrm{Cov}[\Theta_{P|Z_n}] = s^2 I_m$, where $s^2$ is a small variance. This approximation is motivated by studies interpreting Stochastic Gradient Descent (SGD) as a form of approximate Bayesian inference, where the distribution of parameters after training can resemble a Gaussian centered near a mode of a posterior distribution related to the loss function (Mandt et al., 2017).

2. **Validity of Second-Order KL Approximation**: The KL divergence between outputs of models with slightly different parameters can be accurately approximated by a quadratic form involving the Fisher Information Matrix (FIM). This relies on the parameter difference being small, implying $s^2$ must be small.

3. **Effective FIM Constancy in Expectation**: The variations of the FIM $F(\theta')$ for $\theta' \sim \mathcal{N}(\theta_{Z_n}, s^2 I_m)$ around $F(\theta_{Z_n})$ are assumed to average out sufficiently within the expectation required to calculate $\mathcal{C}_{P|Z_n}$. This allows the approximation $\mathcal{C}_{P|Z_n} \approx s^2\mathrm{Tr}(F(\theta_{Z_n}))$.

**Approximation of $\mathcal{C}_P$**   We first consider the conditional inconsistency for a fixed $Z_n$, denoted $\mathcal{C}_{P|Z_n}$, by removing the outer expectation $\mathbb{E}_{Z_n}$:

$$\mathcal{C}_{P|Z_n} = \mathbb{E}_{\theta, \theta' \sim \Theta_{P|Z_n}} \mathbb{E}_x[\mathrm{KL}(f(\theta, x) \| f(\theta', x))]$$

Applying the isotropic covariance posterior assumption, $\theta = \theta_{Z_n} + \delta$ and $\theta' = \theta_{Z_n} + \delta'$, where $\delta, \delta'$ are independent perturbations ($\mathbb{E}[\delta] = \mathbb{E}[\delta'] = 0, \mathrm{Cov}[\delta] = \mathrm{Cov}[\delta'] = s^2 I_m$).

$$\mathcal{C}_{P|Z_n} \approx \mathbb{E}_{\delta, \delta'} \mathbb{E}_x[\mathrm{KL}(f(\theta_{Z_n} + \delta, x) \| f(\theta_{Z_n} + \delta', x))]$$

Using the second-order Taylor expansion for KL divergence taking the expectation over $x$, valid for small $\|\delta - \delta'\|$ (i.e., small $s^2$):

$$\mathbb{E}_x[\mathrm{KL}(f(\theta_{Z_n} + \delta, x) \| f(\theta_{Z_n} + \delta', x))] = \frac{1}{2}(\delta - \delta')^T F(\theta_{Z_n} + \delta')(\delta - \delta') + O(\|\delta\|^3)$$

Let $u = \theta - \theta' = \delta - \delta'$. Since $\delta, \delta'$ are independent, $u \sim \mathcal{N}(0, 2s^2 I_m)$. Substituting this into the expression for $\mathcal{C}_{P|Z_n}$:

$$\begin{aligned}
\mathcal{C}_{P|Z_n} &= \mathbb{E}_u\left[\frac{1}{2}u^T F(\theta')u\right] + O(\|\delta\|^3) \\
&= \mathbb{E}_u\left[\frac{1}{2}u^T F(\theta_{Z_n})u\right] + O(\|\delta\|^3) \quad \text{(FIM Constancy in Expectation Assumption)} \\
&= \frac{1}{2}\mathrm{Tr}(\mathrm{Cov}(u)F(\theta_{Z_n})) + \frac{1}{2}\mathbb{E}[u]^T F(\theta_{Z_n})\mathbb{E}[u] + O(\|\delta\|^3) \\
&= \frac{1}{2}\mathrm{Tr}(2s^2 I_m F(\theta_{Z_n})) + 0 + O(\|\delta\|^3) \quad (\mathbb{E}[u] = 0) \\
&\approx s^2 \mathrm{Tr}(F(\theta_{Z_n}))
\end{aligned}$$

Thus, the conditional inconsistency for a fixed $Z_n$ is approximately proportional to the trace of the FIM evaluated at $\theta_{Z_n}$:

$$\mathcal{C}_{P|Z_n} \approx s^2 \mathrm{Tr}(F(\theta_{Z_n})) \tag{12}$$

The overall inconsistency $\mathcal{C}_P$ is the expectation of this quantity over $Z_n$: $\mathcal{C}_P \approx \mathbb{E}_{Z_n}[s^2 \mathrm{Tr}(F(\theta_{Z_n}))]$.

**Approximation of $S_\rho(\theta_{Z_n})$**   Applying the same second-order KL approximation to the definition of $S_\rho(\theta_{Z_n})$:

$$S_\rho(\theta_{Z_n}) = \max_{\|\delta\| \leq \rho} \frac{1}{2}\delta^\top F(\theta_{Z_n})\delta + O(\|\delta\|^3)$$

The maximum value of the quadratic form $\delta^T A\delta$ for a positive semi-definite matrix $A$ subject to $\|\delta\| \leq \rho$ is achieved when $\delta$ is aligned with the eigenvector corresponding to the largest eigenvalue ($\lambda_{\max}(A)$) and has norm $\rho$. Thus:

$$S_\rho(\theta_{Z_n}) = \frac{1}{2}\rho^2 \lambda_{\max}(F(\theta_{Z_n})) \tag{13}$$

This shows that the local sensitivity $S_\rho$ is approximately proportional to the largest eigenvalue of the FIM.

**Connecting $\mathcal{C}_{P|Z_n}$ and $S_\rho(\theta_{Z_n})$**   For a $m \times m$ positive semi-definite matrix $A$, the relationship between its trace and largest eigenvalue is given by $\frac{1}{m}\mathrm{Tr}(A) \leq \lambda_{\max}(A) \leq \mathrm{Tr}(A)$. Applying this to the FIM $F(\theta_{Z_n})$:

$$\frac{1}{m}\mathrm{Tr}(F(\theta_{Z_n})) \leq \lambda_{\max}(F(\theta_{Z_n})) \leq \mathrm{Tr}(F(\theta_{Z_n}))$$

Substituting this into the approximation for $S_\rho(\theta_{Z_n})$ from Eq. (13):

$$\frac{\rho^2}{2m}\mathrm{Tr}(F(\theta_{Z_n})) \leq S_\rho(\theta_{Z_n}) \leq \frac{\rho^2}{2}\mathrm{Tr}(F(\theta_{Z_n}))$$

Let's assume a plausible connection, for instance, $s^2 = \rho^2/m$. Substituting this into the approximation for $\mathcal{C}_{P|Z_n}$ from Eq. (12), we get $\mathcal{C}_{P|Z_n} \approx \frac{\rho^2}{m}\mathrm{Tr}(F(\theta_{Z_n}))$. Combining this with the bounds for $S_\rho(\theta_{Z_n})$:

$$\frac{1}{2}\left(\frac{\rho^2}{m}\mathrm{Tr}(F(\theta_{Z_n}))\right) \leq S_\rho(\theta_{Z_n}) \leq \frac{m}{2}\left(\frac{\rho^2}{m}\mathrm{Tr}(F(\theta_{Z_n}))\right).$$

This leads to the final approximate relationship between the conditional inconsistency (for a fixed $Z_n$) and the local sensitivity (at the corresponding $\theta_{Z_n}$):

$$\frac{1}{2}\mathcal{C}_{P|Z_n} \leq S_\rho(\theta_{Z_n}) \leq \frac{m}{2}\mathcal{C}_{P|Z_n} \tag{14}$$

This result suggests that, under the stated assumptions, the conditional inconsistency $\mathcal{C}_{P|Z_n}$ and the local sensitivity $S_\rho(\theta_{Z_n})$ are approximately proportional, with the proportionality factor potentially depending on the parameter dimension $m$.

**Anisotropic covariance**    Let $\mathrm{Cov}[\theta_{P|Z_n}] = s^2\Sigma$, where $s^2 = \frac{\rho^2}{m}$. Starting from $\mathcal{C}_{P|Z_n} = \frac{1}{2}\mathrm{Tr}(\Sigma F(\theta_{Z_n}))$,

$$\lambda_{min}(\Sigma)\mathrm{Tr}(F) \leq \mathrm{Tr}(\Sigma F) \leq \lambda_{\max}(\Sigma)\mathrm{Tr}(F)$$

$$\lambda_{min}(\Sigma)\lambda_{\max}(F) \leq \mathrm{Tr}(\Sigma F) \leq \lambda_{\max}(\Sigma)m\lambda_{\max}(F)$$

$$\frac{\rho^2}{2m\lambda_{\max}(\Sigma)}\mathrm{Tr}(\Sigma F) \leq \frac{\rho^2}{2}\lambda_{\max}(F) \leq \frac{\rho^2}{2\lambda_{min}(\Sigma)}\mathrm{Tr}(\Sigma F)$$

$$\frac{1}{\lambda_{\max}(\Sigma)}\mathcal{C}_{P|Z_n} \leq S_\rho(\theta_{Z_n}) \leq \frac{m}{\lambda_{min}(\Sigma)}\mathcal{C}_{P|Z_n}$$

**Practical Considerations: Eigenvalue Spectrum of Neural Networks**    In practice, for deep learning models, the FIM often exhibits a sparse eigenvalue spectrum: many eigenvalues are close to zero, and only a few are significantly large. In such cases:

- The trace $\mathrm{Tr}(F) = \sum \lambda_i$ is dominated by the sum of the few large eigenvalues.

- The ratio $\lambda_{\max}(F)/\mathrm{Tr}(F)$ might be closer to $1/m'$ than $1/d$, where $m' \ll m$ is the "effective rank" or number of dominant eigenvalues.

This implies that the bounds relating $\lambda_{\max}(F)$ and $\mathrm{Tr}(F)$ might be tighter than the general $1/m$ and $1$ factors suggest. Consequently, the relationship between $\mathcal{C}_{P|Z_n}$ (related to trace) and $S_\rho$ (related to max eigenvalue) could be closer to direct proportionality than Eq. (5) indicates, especially if $s^2$ is appropriately related to $\rho^2$.

**Summary and Limitations**    This analysis provides a heuristic argument suggesting a connection between conditional inconsistency $\mathcal{C}_{P|Z_n}$ and local sensitivity $S_\rho(\theta_{Z_n})$. Under assumptions of a Gaussian posterior, small variance $s^2$, validity of second-order KL approximations, local FIM constancy, and a specific link between $s^2$ and $\rho^2$ (e.g., $s^2 = \rho^2/m$), we find that $S_\rho(\theta_{Z_n})$ is approximately proportional to $\mathcal{C}_{P|Z_n}$, potentially up to a factor related to dimension $m$. This connection is mediated by the trace and the maximum eigenvalue of the Fisher Information Matrix. The practical observation of sparse FIM eigenvalues might strengthen this relationship.

## C. Decision boundary of neural networks and principal eigenspace of FIM

To intuitively analyze the role of $\delta_1$ in the training of a neural network, we conducted experiments using a 3-layer fully-connected neural network on two-dimensional synthetic data. The data is generated from a mixture of three Gaussian distributions, a setup analogous to that employed by (Jang et al., 2022) in their investigation of the characteristics of the FIM eigensubspace. Their work demonstrated that perturbing parameters along the principal eigenvectors of the FIM can lead to significant modifications in the decision boundary, such as increasing or decreasing the margins of specific classes.

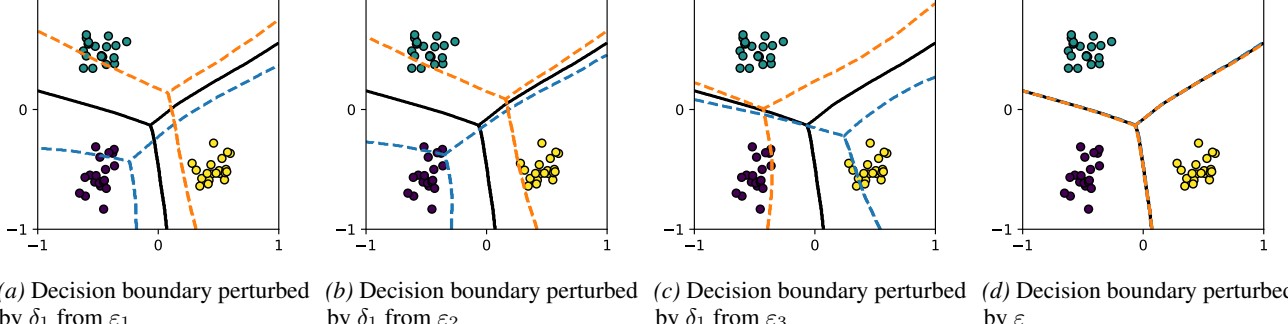

*(a)* Decision boundary perturbed by $\delta_1$ from $\varepsilon_1$    *(b)* Decision boundary perturbed by $\delta_1$ from $\varepsilon_2$    *(c)* Decision boundary perturbed by $\delta_1$ from $\varepsilon_3$    *(d)* Decision boundary perturbed by $\varepsilon$

*Figure 5.* A synthetic classification example. The black, blue, orange lines correspond to decision boundaries of the NN with trained parameter values, and parameter values perturbed by $\delta_1$. Each plot use different noise.

Our investigation focuses on whether $\delta_1$, despite being derived from only a single gradient step (as described in Algorithm 1) and thus influenced by an initial random noise vector $\varepsilon$, still induces substantial changes in the neural network's decision boundary. Figure 5 visualizes these effects. The black lines in each subfigure depict the original decision boundary obtained with the trained parameters $w$. Figure 5 (a-c) show the perturbed decision boundaries (blue and orange lines) when distinct $\pm\delta_1$ with $\rho = 0.5$ is added to $w$. Each of these $\delta_1$ vectors was computed using a different random initialization noise vector, denoted as $\varepsilon_1$, $\varepsilon_2$, and $\varepsilon_3$, respectively. For a direct comparison of the perturbation's nature, Figure 5(d) illustrates the decision boundary perturbed by directly adding the random noise vector $\varepsilon$ to $w$. This vector $\varepsilon$ is sampled from the same distribution as initial vectors (e.g.$\varepsilon_1$) and is scaled to $\|\varepsilon\| = \rho$ the same as $\delta_1$. As observed in Figure 5 (d), direct perturbation with such an arbitrary random noise vector does not meaningfully alter the decision boundary, even when its norm is equivalent to that of the $\delta_1$. This is sharply opposed to the significant changes induced by $\delta_1$ perturbations shown in Figures 5 (a-c), underscoring that the direction derived by Algorithm 1, even in a single step, is substantially more influential than arbitrary noise of the same magnitude. This result intuitively suggests that the perturbation $\delta_1$ with single gradient step is still meaningful and aligned with principal eigenvectors of FIM.

To investigate the alignment between the single-step perturbation vector $\delta_1$ and the principal eigenspace of the FIM, we explicitly compute the FIM and its top three eigenvectors $v_1$, $v_2$, $v_3$, corresponding to the largest eigenvalues $\lambda_1 > \lambda_2 > \lambda_3$. The perturbation $\delta_1$ results from one normalized gradient ascent step on the KL divergence objective, starting from an initial random noise $\varepsilon$. In the language of power iteration, $\delta_1$ before normalization sums the FIM eigenvectors weighted by $\lambda_i \alpha_i$. Formally, write $\varepsilon$ in the eigenbasis of $F(w)$ as $\varepsilon = \sum_i^m \alpha_i v_i$ with $\varepsilon \sim \mathcal{N}(0, \sigma^2 I_m)$; the coefficients $\alpha_i$ are i.i.d. $\mathcal{N}(0, \sigma^2)$ because $\{v_i\}$ form an orthonormal basis.

$$F(\theta)\varepsilon = \sum_i^m \lambda_i v_i v_i^\top \sum_i^m \alpha_i v_i$$
$$= \sum_i^m \lambda_i \alpha_i v_i$$

The cosine similarity between $\delta_1$ and $v_i$ is therefore $\lambda_i \alpha_i$, and the projection onto the principal subspace, $\frac{\|\sum_i^3 \delta_1^\top v_i\|}{\|\delta_1\|}$, equals $\frac{\|\sum_i^3 \alpha_i \lambda_i v_i\|}{\|\delta\|}$.

Figure 6 presents empirical results from this analysis. Panel (a) shows histograms of the absolute cosine similarities between $\delta_1$ (generated from 10,000 different $\varepsilon$ samples) and each of the top three eigenvectors. $\delta_1$ aligns more strongly with $v_1$, the

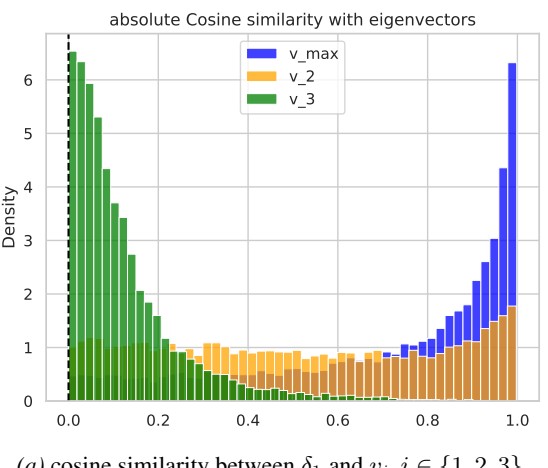
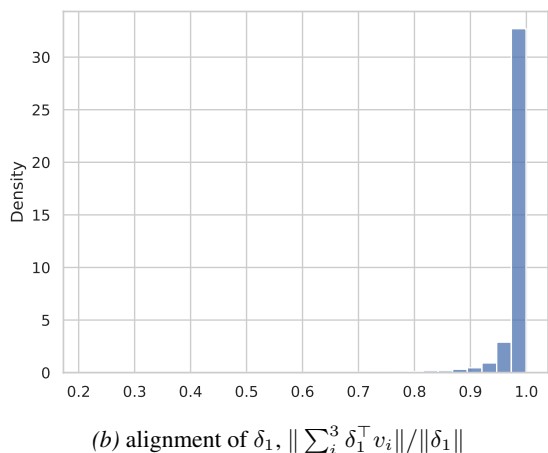

*(a)* cosine similarity between $\delta_1$ and $v_i, i \in \{1, 2, 3\}$      *(b)* alignment of $\delta_1$, $\| \sum_i^3 \delta_1^\top v_i \| / \|\delta_1\|$

*Figure 6.* A synthetic classification example. $\delta_1$ aligns with the top three eigenvectors of the FIM, sampled from 10,000 Gaussian noises $\varepsilon$.

eigenvector of the largest eigenvalue $\lambda_1$, than with $v_2$ or $v_3$. Panel (b) shows the squared norm of the projection of $\delta_1$ onto the top-3 eigenspace; the values concentrate near 1, so $\delta_1$ vectors from different initializations remain largely within this principal subspace. These results support the theoretical expectation that the single-step perturbation $\delta_1$ aligns predominantly with the principal eigenspace of the FIM.

**Accuracy of Algorithm 1 against the exact maximizer.** The analysis above concerns a single step ($K = 1$). We now verify that Algorithm 1 recovers the exact local inconsistency as $K$ grows. On the same 3-layer MLP and 2D Gaussian-mixture setup, we explicitly form $F \in \mathbb{R}^{5393 \times 5393}$ and compare three quantities: the theoretical value $\frac{\rho^2}{2}\lambda_{\max}(F)$ from a full eigendecomposition; the numerical maximum $S_\rho^*$ from an extensive Projected Gradient Ascent (20 multi-starts, 200 steps, initial step size $\rho/8$ with scheduled decay); and our estimate $\hat{S}_\rho$ from Algorithm 1. Algorithm 1 recovers 99.77% of the exact maximum value (mean relative error 0.23%), and its perturbation direction $\delta_{\text{alg1}}$ aligns with the theoretical principal eigenvector $v_1$ at cosine similarity $0.9960 \pm 0.0004$.

To confirm this holds throughout training rather than only at convergence, we track all three quantities over the full optimization, recomputing the exact eigendecomposition every 10 steps. To check that Algorithm 1 behaves correctly as $K$ increases, we run this comparison with $K = 10$. Figure 7 shows that $\hat{S}_\rho$ tracks the theoretical maximum with negligible gap across training, indicating that the low approximation error is stable rather than incidental to a particular checkpoint.

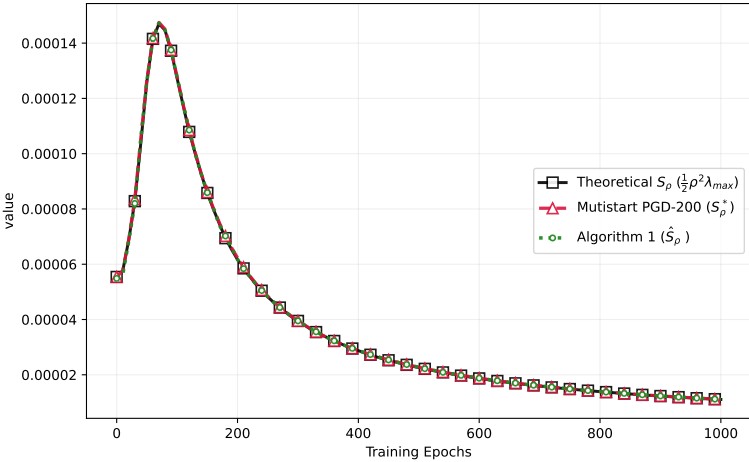

*Figure 7.* Optimization dynamics of the theoretical value $\frac{\rho^2}{2}\lambda_{\max}(F)$, the numerical exact maximum $S_\rho^*$, and the Algorithm 1 estimate $\hat{S}_\rho$ ($K = 10$) over training. The three curves remain tightly overlapped throughout.

# D. Theoretical Justification for Detaching the Perturbation in IAM-D

**Takeaway.** In IAM-D we optimize $L_{\text{IAM-D}}(\theta) = L(\theta) + \beta S_\rho(\theta)$. Under the second-order (quadratic) approximation of the KL divergence, the local inconsistency is well-approximated by the FIM spectral norm:

$$\tilde{S}_\rho(\theta) := \max_{\|\delta\| \le \rho} \frac{1}{2} \delta^\top F(\theta) \delta = \frac{1}{2} \rho^2 \lambda_{\max}(F(\theta)). \tag{15}$$

Crucially, the exact gradient of $\tilde{S}_\rho(\theta)$ can be computed *without differentiating through the maximizer* $\delta^*(\theta)$. Equivalently, applying a stop-gradient (detach) to $\delta^*$ yields the correct outer gradient (envelope theorem / Hellmann–Feynman viewpoint).

## D.1. Quadratic surrogate and its maximizer

Since $F(\theta) \succeq 0$, the maximizer of the Rayleigh quotient is attained at the top eigenvector. Let $v_{\max}(\theta)$ be a unit-norm eigenvector associated with $\lambda_{\max}(F(\theta))$ and define

$$\delta^*(\theta) := \rho\, v_{\max}(\theta). \tag{16}$$

Then $\tilde{S}_\rho(\theta) = \frac{1}{2}(\delta^*)^\top F(\theta) \delta^* = \frac{1}{2}\rho^2 \lambda_{\max}(F(\theta))$.

## D.2. Stop-gradient computes the exact gradient of $\tilde{S}_\rho$

We differentiate the quadratic form $\frac{1}{2}(\delta^*)^\top F(\theta)\delta^*$ w.r.t. $\theta$. For each coordinate $\theta_j$, the chain rule gives

$$\frac{\partial}{\partial \theta_j} \tilde{S}_\rho(\theta) = \underbrace{\frac{1}{2}(\delta^*)^\top \frac{\partial F(\theta)}{\partial \theta_j} \delta^*}_{\text{(A) curvature term}} + \underbrace{(\delta^*)^\top F(\theta) \frac{\partial \delta^*}{\partial \theta_j}}_{\text{(B) perturbation term}}. \tag{17}$$

Term (A) is exactly what is computed when we stop gradients through $\delta^*$.

To show that term (B) vanishes for the exact maximizer, note that $F(\theta)\delta^* = \lambda_{\max}(F(\theta))\delta^*$, hence

$$(\delta^*)^\top F(\theta) \frac{\partial \delta^*}{\partial \theta_j} = \lambda_{\max}(F(\theta))\, (\delta^*)^\top \frac{\partial \delta^*}{\partial \theta_j}. \tag{18}$$

Moreover, $\|\delta^*(\theta)\| = \rho$ is constant by construction, so

$$\frac{\partial}{\partial \theta_j} \|\delta^*(\theta)\|^2 = \frac{\partial}{\partial \theta_j}\big((\delta^*)^\top \delta^*\big) = 2(\delta^*)^\top \frac{\partial \delta^*}{\partial \theta_j} = 0, \tag{19}$$

which implies $(\delta^*)^\top \frac{\partial \delta^*}{\partial \theta_j} = 0$ and therefore term (B) is zero. Plugging back into (17),

$$\nabla_\theta \tilde{S}_\rho(\theta) = \nabla_\theta \Big(\frac{1}{2}\delta^\top F(\theta)\delta\Big)\Big|_{\delta = \delta^*(\theta)} = \frac{1}{2}(\delta^*)^\top (\nabla_\theta F(\theta))\, \delta^*. \tag{20}$$

Finally, when $\lambda_{\max}(F(\theta))$ is simple, the Hellmann–Feynman theorem gives

$$\nabla_\theta \lambda_{\max}(F(\theta)) = v_{\max}(\theta)^\top (\nabla_\theta F(\theta)) v_{\max}(\theta), \tag{21}$$

and thus (20) can be equivalently written as

$$\nabla_\theta \tilde{S}_\rho(\theta) = \frac{1}{2}\rho^2\, \nabla_\theta \lambda_{\max}(F(\theta)). \tag{22}$$

(If $\lambda_{\max}$ has multiplicity $> 1$, the same expression yields a valid subgradient for any unit vector in the top eigenspace.)

## D.3. Implication for IAM-D implementation

In practice, $\delta^*$ is approximated by $K$ steps of power iteration (Algorithm 1), producing $\delta_K$. We implement IAM-D by *stopping gradients through* $\delta_K$ when computing the outer gradient:

$$g_{\text{reg}} := \nabla_\theta \Big( \beta\, \widehat{S}_\rho(\theta;\, \text{sg}(\delta_K)) \Big), \tag{23}$$

where $\text{sg}(\cdot)$ denotes the stop-gradient operator. When $\delta_K \approx \delta^*$, this matches the exact gradient of the quadratic surrogate $\tilde{S}_\rho(\theta)$, while avoiding costly higher-order terms arising from differentiating through the inner maximization.

# E. Extra experiments

### E.1. Analysis of Local Inconsistency Estimation

In this section, we analyze how the estimation of local inconsistency $S_\rho(\theta)$ is affected by approximation choices: the mini-batch size used for perturbation ($m$-sharpness) and the number of gradient ascent steps ($K$). Regarding $m$-sharpness, we follow the protocol introduced for SAM—computing *independent* perturbations on disjoint sub-batches in parallel and *averaging* the perturbed gradients for the update—and replicate this scheme for IAM-S. $m$ indicates the size of disjoint sub-batch.

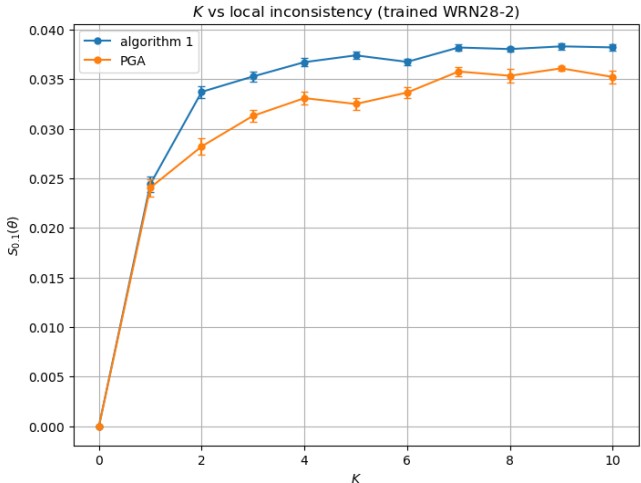

*Figure 8.* Estimated $S_\rho(\theta)$ with respect to $K$ on WRN28-2 (CIFAR-10) using Algorithm 1 vs. Projected Gradient Ascent. Algorithm 1 with $K = 3$ offers a sufficient approximation of the true maximizer.

We investigate the impact of the number of steps $K$ used in Algorithm 1 on model performance. From the perspective of estimating $S_\rho(\theta)$, increasing $K$ naturally yields a more accurate approximation of the worst-case perturbation $\delta^*$ and, consequently, a tighter lower bound on the local inconsistency, as illustrated in Figure 8. This suggests that a more precise estimation of $S_\rho(\theta)$ (i.e., using $K > 1$) during training may lead to better regularization and improved generalization.

To verify this, we conducted an ablation study on $K$ using IAM-D trained on CIFAR-10/100, following the standard hyperparameters described in Appendix F. The results are summarized in Table 4.

As shown in Table 4, we observe a consistent improvement in generalization performance as $K$ increases. Specifically, increasing $K$ from 1 to 3 reduces the test error from 3.28% to 2.99%, although this comes at the cost of increased computational overhead.

Notably, this finding stands in contrast to SAM, where increasing the number of inner maximization steps was reported to have no strong effect on test accuracy for CIFAR-10. While SAM found that a single step was sufficient to obtain a good approximation of the maximizer, our results indicate that for IAM, a more accurate estimation of the local inconsistency via multiple steps ($K > 1$) provides tangible benefits to the final model performance.

*Table 4.* Test error and training cost of IAM with respect to $K$ and $m$ (WRN28-10).

| $K$ | CIFAR-10 | | | CIFAR-100 | | | Running time (s/epoch) |
|---|---|---|---|---|---|---|---|
| | Standard | $m = 32$ | $m = 16$ | Standard | $m = 32$ | $m = 16$ | |
| 1 | 3.28 $\pm$ 0.06 | 3.05 $\pm$ 0.02 | 3.03 $\pm$ 0.02 | 17.16 $\pm$ 0.03 | 16.92 $\pm$ 0.04 | 16.58 $\pm$ 0.05 | 239 (1.0×) |
| 2 | 3.03 $\pm$ 0.02 | 2.85 $\pm$ 0.04 | **2.80** $\pm$ 0.02 | 16.92 $\pm$ 0.04 | 16.08 $\pm$ 0.08 | 15.45 $\pm$ 0.09 | 311 (1.3×) |
| 3 | **2.99** $\pm$ 0.04 | 2.86 $\pm$ 0.01 | 2.91 $\pm$ 0.02 | 16.90 $\pm$ 0.03 | 15.89 $\pm$ 0.01 | 15.34 $\pm$ 0.05 | 378 (1.6×) |
| 5 | **2.98** $\pm$ 0.03 | **2.80** $\pm$ 0.08 | 2.87 $\pm$ 0.01 | **16.62** $\pm$ 0.02 | **15.78** $\pm$ 0.18 | **15.26** $\pm$ 0.01 | 525 (2.2×) |

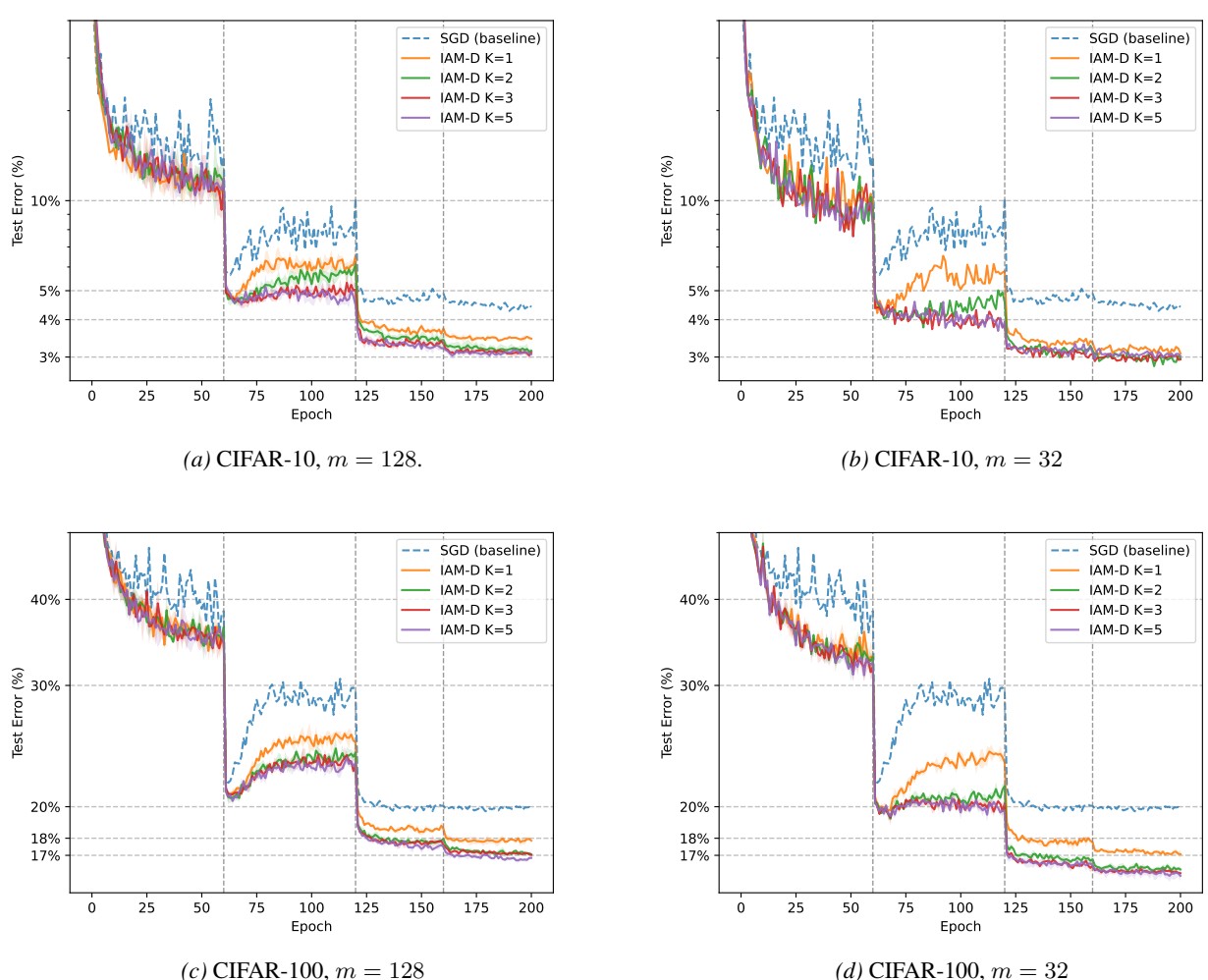

*(a)* CIFAR-10, $m = 128$.

*(b)* CIFAR-10, $m = 32$

*(c)* CIFAR-100, $m = 128$

*(d)* CIFAR-100, $m = 32$

*Figure 9.* The evolution of test error (log-scale) with SGD and IAM-D according to different sub-batch size and K

**m-sharpness in IAM: parallel per-sub-batch perturbations** On CIFAR-10 with a fixed total batch size 256, we split each batch into sub-batch size $m \in \{4, 16, 64, 256\}$, compute perturbation $\delta$ and gradient $\nabla_\theta L(\theta + \delta)$ on each mini-batch and update with the mean of gradients. We sweep $\rho \in \{0.005, 0.01, 0.02, 0.05, 0.1, 0.2, 0.5\}$ and repeat each $(m, \rho)$ condition three times with independent seeds. All other training details (backbone, schedule, preprocessing) are identical to the main IAM-S experiments in the paper.

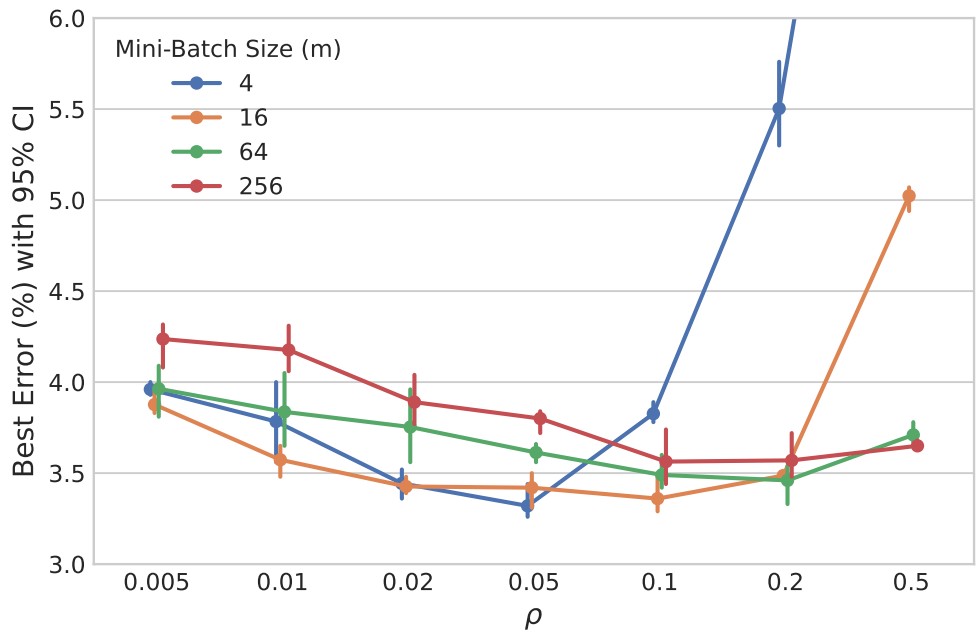

*Figure 10.* Test error as a function of $\rho$ for different values of $m$.

Figure 10 shows that smaller values of $m$ tend to yield models having better generalization ability as observed in (Foret et al., 2021).

### E.2. Supervised learning

*Table 5.* Top-1 and Top-5 error (mean $\pm$ stderr) of ResNet-50 trained on ImageNet with different epoch settings.

| Epoch | Metric | SGD | SAM | IAM-D | IAM-S |
|---|---|---|---|---|---|
| 100 | Top-1 | 23.27 $_{\pm 0.08}$ | 22.69 $_{\pm 0.13}$ | **22.39** $_{\pm 0.12}$ | 22.84 $_{\pm 0.04}$ |
|     | Top-5 | 6.72 $_{\pm 0.03}$ | 6.41 $_{\pm 0.03}$ | **6.24** $_{\pm 0.03}$ | 6.46 $_{\pm 0.06}$ |
| 200 | Top-1 | 22.66 $_{\pm 0.12}$ | 21.80 $_{\pm 0.12}$ | **21.36** $_{\pm 0.06}$ | 21.72 $_{\pm 0.07}$ |
|     | Top-5 | 6.51 $_{\pm 0.07}$ | 5.99 $_{\pm 0.04}$ | **5.70** $_{\pm 0.02}$ | 5.90 $_{\pm 0.02}$ |
| 400 | Top-1 | 22.80 $_{\pm 0.23}$ | 21.45* | **20.73*** | 20.95* |
|     | Top-5 | 6.66 $_{\pm 0.06}$ | 5.95 | **5.53** | 5.61 |

*Results except for SGD at 400 epochs are from a single run.

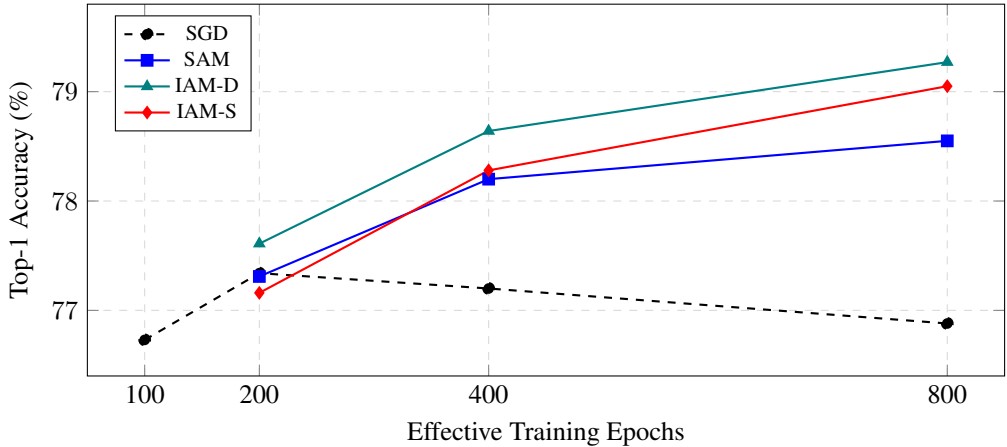

*Figure 11.* **Top-1 Accuracy vs. Effective Training Cost on ImageNet.** Since SAM and IAM variants require two backward passes per step, their effective epochs are doubled compared to SGD. Results of SGD at 800 epochs are from a single run. **IAM-D** consistently achieves the best accuracy-efficiency trade-off across all training budgets.

Table 5 reports Top-1/Top-5 error of ResNet-50 on ImageNet across 100, 200, and 400 training epochs. Consistent with the observation of Foret et al. (2021) that SGD tends to overfit as training is extended from 200 to 400 epochs while SAM continues to improve, we find that SGD's Top-1 error stagnates (and even slightly degrades) from 22.66% at 200 epochs to 22.80% at 400 epochs, whereas SAM further reduces error to 21.45%. Both IAM variants exhibit the same favorable scaling behavior as SAM: IAM-D improves from 21.36% to 20.73% and IAM-S from 21.72% to 20.95% as the training budget doubles. Notably, IAM-D consistently achieves the lowest error across all epoch settings, outperforming SAM by 0.30%, 0.44%, and 0.72% at 100, 200, and 400 epochs, respectively. This widening gap suggests that the regularization induced by local inconsistency becomes increasingly beneficial under longer training, where overfitting pressure is stronger.

A natural concern with IAM and SAM-style methods is the additional computational cost introduced by the inner perturbation step, which requires an extra forward-backward pass per update. To assess whether the observed gains justify this overhead, we compare optimizers on equal footing in terms of *effective training epochs*, defined as the number of gradient evaluations normalized by that of one SGD epoch. Under this metric, one epoch of SAM, IAM-D, or IAM-S costs twice as much as one epoch of SGD. Figure 11 plots Top-1 accuracy against effective training epochs for all methods. Even at matched compute, IAM-D dominates the best accuracy at every compute budget: at 400 effective epochs, IAM-D attains 78.64% accuracy, surpassing SGD trained for 800 effective epochs (76.88%) by 1.76% and SAM at the same budget (78.20%) by 0.44%. The gap persists at 800 effective epochs, where IAM-D reaches 79.27% versus 78.55% for SAM. IAM-S exhibits a similar advantage over SAM, while SGD's accuracy plateaus and eventually deteriorates due to overfitting. These results indicate that the extra gradient computation incurred by IAM is not merely a fixed-cost tax but yields a strictly better accuracy-compute trade-off than both vanilla SGD and SAM across the entire range of training budgets we examined.

## E.3. ViT

*Table 6.* Test error $\pm$ stderr of SGD, SAM, and IAM-D when fine-tuning ViT-S/16 on CIFAR-10.

| Method | Test error |
|--------|------------|
| SGD | $1.86_{\pm 0.01}$ |
| SAM | $1.56_{\pm 0.01}$ |
| IAM-D | $1.52_{\pm 0.02}$ |

To demonstrate the versatility of IAM beyond CNN architectures, we conducted additional fine-tuning experiments on ViT-S/16 pre-trained on ImageNet-1K using the CIFAR-10 dataset. We compared IAM-D against SGD and SAM.

We fine-tuned the models for 10,000 steps with a batch size of 128, with base optimizer SGD. Gradient clipping with max norm $= 1.0$ is applied. The initial learning rate was set to 0.01 with a linear decay schedule after 500 warmup steps. For perturbation magnitude $\rho$, we used $\rho = 0.05$ for SAM and $\rho = 0.1$ for IAM-D.

*Table 7.* Test error (mean $\pm$ stderr) of ViT-T/4 trained 200 epochs on CIFAR-10 and CIFAR-100.

| Optimizer | CIFAR-10 | CIFAR-100 |
|-----------|----------|-----------|
| Adam-W | $15.29_{\pm 0.50}$ | $43.73_{\pm 0.19}$ |
| SAM | $14.39_{\pm 0.31}$ | $42.45_{\pm 0.35}$ |
| ASAM | $14.60_{\pm 0.25}$ | $41.78_{\pm 0.33}$ |
| IAM-D | $\mathbf{13.93}_{\pm 0.17}$ | $\mathbf{40.83}_{\pm 0.42}$ |
| IAM-S | $14.37_{\pm 0.19}$ | $42.25_{\pm 0.42}$ |

We also train ViT-Tiny/4 on CIFAR-{10,100}, with Adam-W as default optimizer. The initial learning rate was set to 0.003 with a linear decay schedule after 3000 warm-up steps. Other settings are the same as main experiments.

*Table 8.* Top-1 and Top-5 error (mean $\pm$ stderr) of ViT-S/32 trained 300 epochs on ImageNet.

| | Top-1 | Top-5 |
|--------|-------|-------|
| Adam-W | $33.81_{\pm 0.07}$ | $13.96_{\pm 0.07}$ |
| SAM | $32.24_{\pm 0.17}$ | $12.84_{\pm 0.07}$ |
| IAM-D | $31.03_{\pm 0.10}$ | $11.95_{\pm 0.08}$ |
| IAM-S | $\mathbf{30.09}_{\pm 0.30}$ | $\mathbf{11.38}_{\pm 0.19}$ |

In addition, we train ViT-S/32 on ImageNet, with Adam-W as default optimizer. The initial learning rate was set to 0.0075 with a linear decay schedule after 40000 warm-up steps. We use the same $\rho$ values for SAM, IAM-D and IAM-S as in ResNet-50 experiments.

IAM-D is consistently competitive with SAM on the transformer-based architecture, confirming that our proposed local inconsistency measure is effective across different model inductive biases.

*Table 9.* Test error (mean $\pm$ stderr) of MLP-Mixer-T/4 trained 200 epochs on CIFAR-10 and CIFAR-100.

| Optimizer | CIFAR-10 | CIFAR-100 |
|-----------|----------|-----------|
| Adam-W | $15.81_{\pm 0.45}$ | $43.11_{\pm 0.26}$ |
| SAM | $15.35_{\pm 0.25}$ | $41.26_{\pm 0.60}$ |
| ASAM | $16.01_{\pm 0.31}$ | $43.52_{\pm 0.25}$ |
| IAM-D | $\mathbf{14.63}_{\pm 0.40}$ | $41.51_{\pm 0.32}$ |
| IAM-S | $15.03_{\pm 0.26}$ | $\mathbf{38.39}_{\pm 0.31}$ |

### E.4. Semi-supervised learning

If we restrict SAM only to the **labeled** loss to avoid instability, we only minimize sharpness for the very small subset of labeled data (e.g., 250 samples). This fails to regularize the global landscape. To confirm this, we ran "FixMatch + SAM" on CIFAR-10 (250 labels).

*Table 10.* Test Error (mean ± stderr) on CIFAR-10 with 250 labels using a WRN-28-2 model.

| Method | Test error |
|---|---|
| FixMatch | $6.26_{\pm 0.39}$ |
| FixMatch + SAM | $9.90_{\pm 0.74}$ |
| FixMatch + IAM-D | $\mathbf{5.30}_{\pm 0.08}$ |

This is significantly worse than FixMatch + SGD (6.26 %) and FixMatch + IAM-D (5.30 %). This failure case underscores the strength of IAM-D: it calculates inconsistency on unlabeled data without relying on potentially incorrect pseudo-labels, making it naturally superior for SSL.

### E.5. Self-supervised learning

We extend the SSL experiment in Section 5.3 to CIFAR-100 using the same SimCLR + IAM-D setup described in Appendix, with the encoder evaluated via linear probing. We use $\beta = 1.0, \rho = 0.1$, identical to the CIFAR-10 SSL setting. As shown in

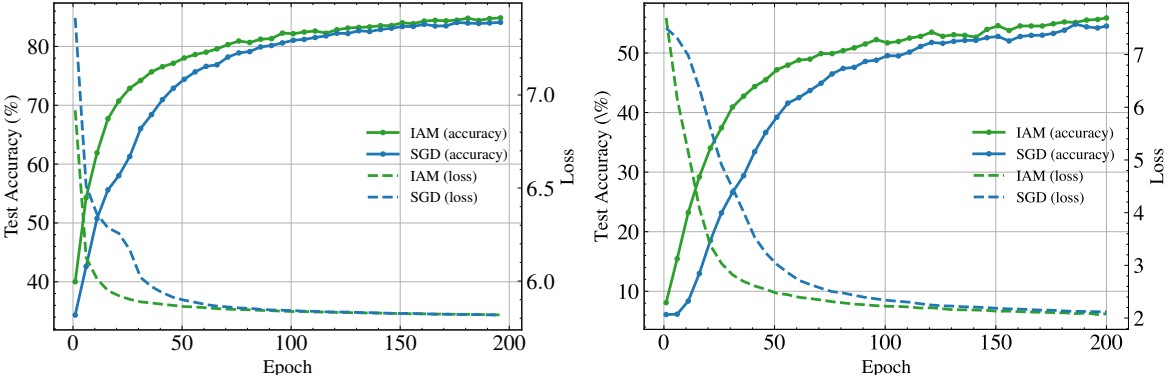

*Figure 12.* Test accuracy on linear probe and SimCLR training loss for ResNet-18 on CIFAR-{10, 100}, comparing SimCLR trained with SGD (SimCLR-SGD) versus SimCLR with IAM-D (SimCLR-IAM).

Figure 12, SimCLR-IAM consistently achieves higher linear probe accuracy than SimCLR-SGD on CIFAR-100, mirroring the trend observed on CIFAR-10. This confirms that controlling local inconsistency benefits self-supervised representation learning regardless of the number of classes.

# F. Experimental Details

**Practical Considerations in estimating $S_\rho(\theta)$**

- **Computational Efficiency:** Calculating the FIM explicitly and performing eigenvalue decomposition is computationally expensive ($O(m^2)$) or worse, where $m$ is the number of parameters). Algorithm 1 avoids this by requiring only $K$ gradient computations (forward and backward passes) per estimation, making its computational cost approximately $O(mK)$, which is significantly more feasible for large networks.

- **Number of Steps (K):** Empirical studies on neural network Hessians and FIMs suggest that the eigenspectrum is often dominated by a large dominant eigenvalue. Thus, the Power Iteration method can converge quickly to the dominant eigenvector. In practice, using a small number of steps, often just $K = 3$, is found to be sufficient to get a reasonable estimate of the maximizing direction. This makes the computation highly efficient.

- **Averaging to reduce variance from initialization:** The estimate of $S_\rho(\theta)$ obtained from Algorithm 1 depends on the random initialization $\delta_0$ when $K = 1$. To obtain a more stable estimate, we compute the metric multiple times (e.g., 10 times) with different random initializations for $\delta_0$ and report the average value: $\mathbb{E}_{\delta_0}[\text{Estimate from Alg 1}]$.

**Infrastructure**    Experiments are implemented in PyTorch 2.5.1 and executed on NVIDIA A40, A100 and L4 GPUs.

## F.1. Experimental details for Figure 1 (Section 4.6)

We trained 6CNN and WRN28-2 using SGD to investigate the relationship between generalization gap and local inconsistency. For 6CNN, each hyperparameter combination was run with 5 independent random seeds to assess variability. $\text{Tr}(H)$, $\lambda_{\max}(H)$ and $S_\rho$ were computed on a 5,000-sample unlabeled held-out set.

*Table 11.* Hyperparameters used for 6CNN and WRN28-2 on CIFAR-10.

| Hyperparameter | 6CNN | WRN28-2 |
|---|---|---|
| Dataset | CIFAR-10 | CIFAR-10 |
| Training data size | 45K | 45K |
| Initial learning rate | {0.001, 0.002, 0.005, 0.01, 0.02, 0.05} | {0.1, 0.03, 0.01} |
| Batch size | {32, 64, 128, 256, 512} | {32, 64, 128, 256, 512} |
| Weight decay | {0.0, $10^{-4}$, $5 \times 10^{-4}$, $10^{-3}$} | {0.0, $10^{-4}$, $5 \times 10^{-4}$} |
| Learning rate scheduling | constant | {cosine annealing, multi-step} |
| Data augmentation | False | {True, False} |
| Label smoothing | – | – |
| Epochs | until convergence ($< 400$) | {150, 200, 300} |
| $K$ | 3 | 1 |

## F.2. Image classification

Each reported metric is the mean $\pm$ standard error computed over minimum test error from three independent runs.

**Dataset.**    We evaluate on **CIFAR-10** (50,000 training, 10,000 test images), **CIFAR-100** (50,000 training, 10,000 test images), Fashion-MNIST, and SVHN (no additional datasets). CIFAR-10, CIFAR-100, and SVHN are resized to $32 \times 32$ and preprocessed with RandomCrop(32, padding= 4). Fashion-MNIST is preprocessed with RandomCrop(28, padding= 4). Below are applied augmentations in common:

- *RandomHorizontalFlip*($p = 0.5$), and

- *Normalization* using the official mean and standard deviation.

No additional augmentation such as Cutout or Mixup is applied.

**Optimization.** The models are trained for **200 epochs** with mini-batch size **128**. We use SGD with momentum 0.9, weight decay $5 \times 10^{-4}$ as an optimizer, and a multistep learning rate schedule that decays the initial rate 0.1 (0.01 for SVHN) by 0.2 at epochs 60, 120, and 160. We report the best score achieved by each SGD training run across either the standard epochs or the doubled epochs.

**Hyperparameters.** For image classification tasks, $\beta, \rho$ are tuned via grid search over $\beta \in \{0.1, 1.0, 5.0, 10.0, 20.0\}, \rho \in \{0.01, 0.05, 0.1, 0.5, 1.0\}$ with validation split using 10% of the training dataset. As seen in Figure 13, the best pairs are $(1.0, 0.1)$ for CIFAR-10 and $(10.0, 0.1)$ for CIFAR-100. For both datasets, $\beta$ and $\rho$ had a trade-off relation.

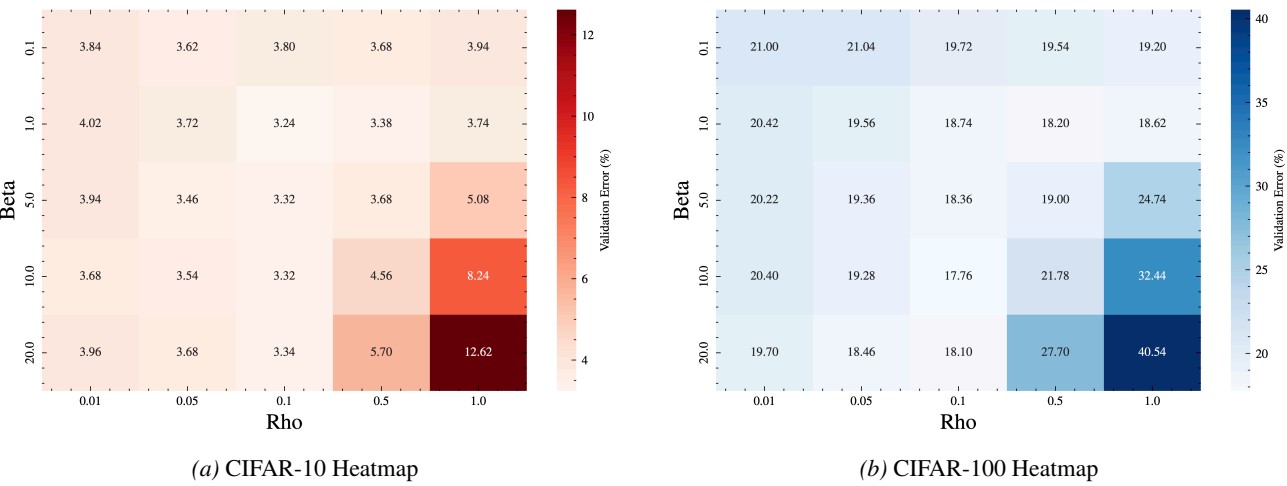

*(a)* CIFAR-10 Heatmap

*(b)* CIFAR-100 Heatmap

*Figure 13.* Test error heatmap of IAM-D.

**Loss function.** Cross-entropy with label smoothing ($\alpha = 0.1$) is used for all methods.

### F.3. Semi-supervised learning

In semi-supervised learning experiment, we shared most of the settings with image classification. Each reported metric is computed over minimum test error from three independent runs. Experiments with FixMatch are stated in a separate section.

**Optimization.** Models are trained for **200 epochs** without learning rate scheduling.

**Hyperparameters.** We used $\beta = 1.0$ and $\rho = 0.1$ for CIFAR-10 and $\beta = 10.0$ and $\rho = 0.1$ for CIFAR-100. SAM is also trained with $\rho = 0.1$. The batch size 128 is used for labeled data and 384 for unlabeled data.

**FixMatch.** We followed the reported FixMatch settings. WRN-28-2 for CIFAR-10, WRN-28-8 for CIFAR-100 are trained for $2^{20}$ **iterations** with SGD as the base optimizer using the learning rate 0.03, momentum 0.9, weight decay $5e - 4$, with cosine learning rate scheduling. For IAM-D, $\rho = 0.01, \beta = 1.0$ is applied for CIFAR-10 and $\rho = 0.05, \beta = 1.0$ is applied for CIFAR-100. The batch size for the labeled data was 64, and for unlabeled data was 448. We applied EMA with decay 0.99.

### F.4. Self-supervised learning

Each reported metric is the mean **test accuracy** obtained from three independent runs.

**Dataset.** We use the CIFAR-10, CIFAR-100 benchmark. All images are resized to $32 \times 32$ and augmented with the SimCLR(Chen et al., 2020) pipeline:

- *RandomResizedCrop*(32, scale=(0.4, 1.0)),

- *RandomHorizontalFlip*($p = 0.5$),

- *ColorJitter*$(0.4, 0.4, 0.2, 0.1)$ with probability $0.8$,

- *RandomGrayscale*$(p{=}0.2)$, and

- *Normalization* using the official mean and standard deviation.

**Encoder & Projection Head.**    We adopt a **ResNet-18** backbone with the first convolution modified to $3{\times}3$ layer with stride $= 1$ and the max-pool removed. The projector is a two-layer MLP (hidden size $512$, output size $128$) with ReLU activation.

**Optimization.**    Models are trained for **200 epochs** with mini-batch size **1024**. We use SGD (momentum 0.9, weight decay $1{\times}10^{-4}$) and a cosine-annealing learning-rate schedule starting at $1.0$ after a 10-epoch warm-up.

**Contrastive Loss.**    The NT-Xent loss is computed with temperature $\tau = 0.5$.

**IAM Hyperparameters.**    We set the inconsistency weight $\beta{=}1.0$, neighborhood radius $\rho{=}0.1$, and noise-scale $3.0$ (Gaussian initialization). The local inconsistency is computed between projection head outputs with temperature $\tau{=}0.5$.

**Stability Heuristics.**    The stability heuristics are identical to those used in the image classification setting.

**Linear Evaluation.**    After every 5 epochs (and at the final epoch), a frozen encoder is evaluated via a linear probe trained for 20 epochs with AdamW optimizer on the full training set (batch size 1024). The reported metric is the probe's test accuracy.

