# OpenReview forum: "Inconsistency-Aware Minimization: Improving Generalization with Unlabeled Data"
_ICML.cc/2026/Conference — ICML 2026 regular_

### Official Review · Reviewer_6RSA · 2026-03-01

**Soundness:** 2
**Presentation:** 3
**Significance:** 2
**Originality:** 3
**Overall Recommendation:** 4
**Confidence:** 4

**Summary:**

This paper introduces "local inconsistency", a novel information-geometric metric designed to estimate the generalization gap. The metric is defined as the worst-case KL divergence between a model's output and its perturbed counterpart and linked to the Fisher Information Matrix and Hessian. This metric can be computed without explicit labels, allowing it to be effectively applied to unlabeled data. Based on this measure, the authors propose Inconsistency-Aware Minimization (IAM), an optimization method that incorporates local inconsistency into the training objective. Experiments demonstrate that IAM achieves competitive performance in supervised learning settings and uniquely improves results in semi-supervised and self-supervised learning.

**Compliance With Llm Reviewing Policy:**

Affirmed.

**Final Justification:**

The paper is well-structured and clearly written, introducing a novel perspective on generalization. The authors’ rebuttal has adequately addressed my main concerns on completeness. Therefore the final score is 4.

**Key Questions For Authors:**

1. How sensitive is the performance of IAM to the choice of the radius $\rho$ and the regularization weight $\beta$? Does the method require extensive dataset-specific tuning compared to standard SAM?

2. In Algorithm 1, the perturbation $\delta$ is initialized randomly. Does this random initialization lead to high variance in the training dynamics?

3. You state that $K=1$ (single power iteration step) is sufficient for performance, but does this choice actually estimate the "worst-case" perturbation defined in Eq. (3)? Have you conducted ablation studies with larger $K$ to see if a more accurate estimation of local inconsistency yields better generalization?

**Limitations:**

yes

**Strengths And Weaknesses:**

**Strengths:**

1. The paper introduces a novel perspective on generalization by proposing "local inconsistency," a metric derived from information geometry that uniquely operates without explicit labels. This label-agnostic property is a significant contribution, as it allows the method to address semi-supervised and self-supervised learning.

2. The paper is well-structured and clearly written. The progression from the definition of local inconsistency to its theoretical connections and to the proposed algorithm is logical, making the complex information-geometric concepts accessible to the reader.

**Weaknesses:**

1. The practical implementation relies on an approximation of the local inconsistency metric by solving a maximization problem over the parameter space. But the algorithm uses only a single step of gradient ascent. There lacks analysis of the approximation error introduced by this "single-step" approach, raising questions about whether the algorithm is truly minimizing the defined metric or just a noisy proxy.

2. The theoretical analysis relies heavily on a second-order Taylor expansion to link the KL divergence to the FIM. However, this approximation is only valid for small perturbations. The paper does not justify whether the hyperparameters used in practice fall within the regime where this approximation holds.

3. The performance improvements in standard supervised learning are marginal. Given that the method introduces additional computational complexity, its impact for standard tasks is limited.

4. The experimental benchmarks are limited in scope. The semi-supervised and self-supervised experiments are restricted to CIFAR-10/100, which are relatively small-scale datasets. Without testing on larger-scale datasets, it is difficult to conclude whether the method scales effectively.

5. The experimental evaluation lacks comparisons with recent relevant works, for instance, SAGM (Sharpness-Aware Gradient Matching). As for the self-supervised experiments, the paper utilize SimCLR, without comparing against more recent frameworks like MAE (Masked Autoencoders).

---

> ### Author Rebuttal · Authors · 2026-03-31
>
> > W1, Q2, Q3. The concern about $K=1$ and the question regarding ablations with $K>1$
>
> We already provide an ablation study on $K$ in the paper and explicitly mention it in the main text. As shown in our ablation study (Appendix E.1, Table 4), more accurate estimation of local inconsistency ($K>1$) consistently improves generalization. However, because $K=1$ already achieves performance comparable to SAM, we adopt it in our main experiments specifically to match SAM's computational budget (one additional gradient computation) for a fair baseline comparison.
>
> Furthermore, $K=1$ is not an arbitrary noisy proxy; it functions as a step of power iteration. Because the Fisher Information Matrix (FIM) of deep neural networks exhibits a highly skewed eigenspectrum dominated by a few large eigenvalues, a single gradient ascent step starting from random noise $ϵ$ amplifies the principal components, expressed as $KL(f(θ)||f(θ + δ_1)) = ρ^2\frac{ϵ^⊤F^3ϵ}{||Fϵ||^2} \propto ∑_i λ_i^3 \alpha_i^2$. Empirical evidence in the paper supports this:
>
> - Appendix C & Figure 5: The single-step perturbation $δ_1$ aligns with the principal eigenspace of the FIM and meaningfully alters the decision boundary, whereas isotropic random noise of the exact same magnitude does not.
>
> - Section 5.2 & Figure 3: Optimizing IAM with $K=1$ successfully and consistently suppresses true local inconsistency across the entire training trajectory compared to SGD.
>
> Formally, letting $v_1$ be the FIM's principal eigenvector, the KL divergence approximation error after $K$ steps splits into a $K$-dependent optimization error and a $ρ$-dependent residual:$$KL(f(θ) || f(θ + ρ v_1)) - KL(f(θ) || f(θ + δ_K)) \le O(ρ^2 e^{-K}) + O(ρ^3)$$This demonstrates the massive principal eigengap rapidly suppresses error even with $K=1$.
>
> Regarding variance (Q2): Random initialization does not induce high variance. Because the KL divergence gradient at $\delta=0$ is exactly zero, $\epsilon$ acts strictly as a minor symmetry-breaking perturbation. Empirically, IAM's standard errors are comparable to or lower than standard baselines (e.g., ImageNet Top-1: IAM-D 21.36 ± 0.06 vs. SGD 22.66 ± 0.12).
>
> ---
> > W2. Validity of the second-order Taylor approximation
>
> Our perturbation radii ($ρ \in \{0.05, 0.1, 0.2, 0.5\}$, typically $0.05$ or $0.1$) are sufficiently small to ensure the approximation's validity. For our models (e.g., WRN28-10 with >25M parameters), the element-wise magnitude of an $\ell_2$-bounded perturbation is $\approx ρ/\sqrt{m}$, which is extremely small ($\approx 10^{-4}$ even for $ρ=0.5$). Prominent theoretical works on sharpness-aware methods (e.g., Luo et al., 2024; Long and Bartlett, 2024) consistently leverage identical second-order expansions.
>
> ---
> >W3 Concern about improvements & computational cost
>
> IAM provides a clear trade-off between computational cost and accuracy. We evaluate this under a strict cost-normalized view where SAM and IAM epochs count twice due to their two backward passes. Under this parity, IAM-D demonstrates a superior accuracy-efficiency trade-off on ImageNet across all compute ranges (App. E.2) and substantially outperforms SAM on CIFAR-100 when given an extended budget ($K=5$).
>
> |Effective Epochs|SGD|SAM|IAM-D|
> |-|-|-|-|
> |200|22.66|22.69|$22.39$|
> |400|22.80|21.80|$21.36$|
> |800*|23.12|21.45|$20.73$|
>
> ||SGD|SAM|IAM-D| IAM-D (K=5)|
> |-|-|-|-|-|
> |cifar-100|19.17|17.63|17.16|$15.26$|
>
> ---
> >W4 Limited scope & larger-scale datasets
>
> Extending our semi-supervised evaluation to STL-10 (96x96 resolution) with a WRN-37-2 model, IAM-D achieves **36.94** $\pm$ 0.48% test error, substantially outperforming the SGD baseline (40.74 $\pm$ 0.35%). To further establish scalability, we are conducting ViT-B/16 fine-tuning following the Unified SSL Benchmark and will provide these results as soon as possible.
>
> ---
> >W5 Comparisons with recent works
>
> Our primary objective is not to exhaustively compare against label-dependent SAM variants, but to establish IAM as a **label-agnostic regularization principle**. The main novelty of our method is that, unlike SAM-style methods, IAM requires no labels to compute the regularizer and naturally extends to semi- and self-supervised learning. To further address your concern, we will provide additional results evaluating IAM within the framework of Lejepa (2025)
>
> ---
> >Q1 Sensitivity to $ρ$ and $β$
>
> IAM does not require extensive tuning. Our sensitivity analysis (App. F.2 and Fig. 9) shows a clear $β$-$ρ$ trade-off with stable optimal regions and competitive performance across a broad range of $ρ$. This sensitivity is entirely comparable to standard SAM-style methods (e.g., Foret et al., 2021, Fig. 3).
>
>
> ---
> reference
>
>
> Long, P. M., & Bartlett, P. L. (2024). Sharpness-Aware Minimization and the Edge of Stability. Journal of Machine Learning Research (JMLR), 25(179), 1-20.

---

> > ### Author Rebuttal · Reviewer_6RSA · 2026-04-01
> >
> > Thank the reviewer for the comprehensive rebuttal. The authors’ rebuttal has adequately addressed my main concerns. Overall, I am satisfied that the major issues I raised have been resolved to a reasonable extent, and I will increase my score to 4.

---

> > > ### Author Response · Authors · 2026-04-08
> > >
> > > Thank you for taking the time to read our rebuttal and for your constructive engagement. We are very glad to hear that our response has adequately addressed your main concerns. We sincerely appreciate your valuable feedback and your kind reconsideration of our work.
> > >
> > > Best regards,
> > > The Authors

---

### Official Review · Reviewer_VTWQ · 2026-03-05

**Soundness:** 3
**Presentation:** 3
**Significance:** 3
**Originality:** 3
**Overall Recommendation:** 4
**Confidence:** 3

**Summary:**

This paper develops a measure called local inconsistency to predict the generalization gap and proposes two optimization frameworks (IAM-D and IAM-S) that incorporate this measure to improve the generalization of deep learning models. Local inconsistency bridges two lines of work, namely, sharpness aware optimization and output-based measures such as inconsistency that are known to correlate with the generalization gap. It presents some beneficial features such as computability from a single trained model, differentiability and independence of labeled data. The effectiveness of the proposed framework is illustrated with synthetic experiments and benchmarks.

**Compliance With Llm Reviewing Policy:**

Affirmed.

**Final Justification:**

I believe this paper has no major issues in its presentation or significance, the latter being supported by the experimental results. The combination of sharpness-aware optimization and the prediction of the generalization gap from inconsistency measures provides appealing features that, to the best of my knowledge, have not been present in those research areas before. This fairly supports the originality of the paper.

My main concern was the soundness of the paper, in particular whether each argument is sufficiently justified.

The technical development of the paper is as follows:

1) It introduces a new metric, called local inconsistency.
2) It connects this metric to several important mathematical concepts.
3) Because direct computation is intractable, it proposes an estimator of local inconsistency (Algorithm 1).
4) It presents experiments based on this approximation.

In my view, the transition to step 3 is somewhat weak. Since all experiments were conducted only in the tractable approximated setting, it remains unclear which properties of the original local inconsistency are actually responsible for the observed results. As a result, the empirical findings may reflect the effectiveness of Algorithm 1 rather than that of local inconsistency itself. An illustrative synthetic experiment, in which local inconsistency could be computed more directly, would have been helpful in addressing this concern. The authors did not sufficiently resolve this point in the rebuttal. Such clarification would also have been valuable for subsequent work as it would help researchers evaluate the method more precisely and make more informed algorithmic design choices for further improvement. The current paper leaves these points unanswered, and the proposed measure remains somewhat of a black box in my view.

Therefore, I keep my score of weak accept.

**Key Questions For Authors:**

Q1. Could the authors identify a problem setting in which the local inconsistency (Eq. (3)) can be computed exactly (i.e., without approximation), thereby enabling clearer visualization of the impact of each approximation, and discuss these effects?

Q2. Does local inconsistency perform better in terms of predictive performance over other metrics such as disagreement and inconsistency?

Q3. Can local inconsistency be extended to more general problem settings beyond classification such as probabilistic/Bayesian frameworks where probability distributions $p_\theta$ are optimized instead of $f(\cdot;\theta)$?

**Limitations:**

The limitations of the proposed approach are not discussed explicitly, although they are implied throughout the paper. A dedicated section summarizing and discussing these limitations would be appreciated. In particular, the intractability of directly computing the proposed measure constitutes a major limitation, which the authors address through a heuristic approximation. Making this point more explicit would strengthen the clarity and balance of the presentation.

**Strengths And Weaknesses:**

(Strengths)

S1. The rationale behind the construction of local inconsistency is well presented and motivated. The combination of ideas of sharpness-aware optimization and the prediction of generalization gap from inconsistency measures leads to desirable features that were not present in those research areas.

S2. Those features are carefully discussed in the paper. Notably, the computation of local inconsistency only relies on unlabeled data, which enables to extend this measure to semi-/self-supervised learning, which is not straightforward for label-constrained sharpness-aware methods

S3. The paper is well-written and presented. The connection of local inconsistency with the Fisher information matrix and loss Hessian strengthens the theoretical justification of this work.

S4. Experiments with IAM-D and IAM-S clearly demonstrate their effectiveness compared to sharpness-aware optimization methods.

(Weaknesses)

W1. The authors primarily focus on classification tasks, and the generalizability of the proposed method to other scenarios remains unclear.

W2. The proposed measure is computationally intractable and requires some heuristic approximations which are provided in Algorithm 1. The impact of such approximations should be discussed more carefully (by extending Appendix E. or in the main paper) before using them to assert the overall effectiveness of the proposed approach through experiments that only use the approximated method. For example, the search domain is restricted from $\|\delta\|<=\gamma$ to $\|\delta\|=\gamma$, which may influence the tightness or validity of the results.

W3. Experiments of the paper primarily investigates optimization procedures for improving the generalization of deep learning models. Since local inconsistency also builds on the inconsistency metric of Johnson et al. (2023), the authors should  additionally compare its predictive performance against this metric and discuss the results for further insights (e.g., by extending the experiment of Section 4.6).

W4. (minor) For better readability, the notation in the main paper should be self-contained. Accordingly, several symbols in Theorem 4.1 would benefit from clarification, such as $\epsilon_R$, $L_\mathcal{D}$... Moreover, in line 195 (left), the authors mention $L_D$, which I guess should be L_\mathcal{D}.

---

> ### Author Rebuttal · Authors · 2026-03-31
>
> We thank the reviewer for the positive assessment of our motivation, theory, and empirical results. In the revision, we will update Theorem 4.1 to make the notation fully self-contained and add a dedicated Limitations section to explicitly discuss the scope and impact of our approximations.
>
>
> > Regarding W1 / Q3 (generality beyond classification).
>
> Local inconsistency naturally extends to any probabilistic predictive distribution, not just classification. Because the metric **is defined via the KL divergence between model outputs**, it directly applies to other domains—such as regression via Gaussian likelihoods (equivalent to MSE up to scaling)—without requiring discrete labels. Furthermore, our experiments integrating IAM into self-supervised (SimCLR) frameworks already shows its practical efficacy well beyond standard supervised classification. We will add discussion of this in the revised paper.
>
> ---
> >W2. The proposed measure is computationally intractable and requires some heuristic approximations which are provided in Algorithm 1. The impact of such approximations should be discussed more carefully (by extending Appendix E. or in the main paper) before using them to assert the overall effectiveness of the proposed approach through experiments that only use the approximated method. For example, the search domain is restricted from $||\delta|| \le \rho$ to $||\delta|| = \rho$, which may influence the tightness or validity of the results.
>
> While exact parameter-space maximization is intractable—a standard challenge shared by SAM-style min-max objectives—Algorithm 1 is a principled mathematical derivation rather than an ad hoc heuristic.
>
> - **Boundary Search** ($||\delta|| \le \rho$ vs. $||\delta|| = \rho$): Under the local quadratic expansion, maximizing the objective over the Euclidean ball is equivalent to maximizing it over its boundary, since the maximum of the positive semidefinite quadratic form is attained at $||δ||=ρ$. Therefore, the **boundary optimum is also the optimum over the original ball-constrained problem**.
>
> - **Visualization of Approximations** (Q1): Appendix C explicitly isolates and visualizes the effect of this approximation. Using a controlled synthetic MLP, we demonstrate that a single-step approximation ($\delta_1$) successfully aligns with the FIM's principal eigenspace and meaningfully shifts the decision boundary, unlike isotropic random noise of the same norm.
>
> - **Computational Budget**: We deliberately default to $K=1$ to guarantee a fair, apples-to-apples comparison with SAM and ASAM, ensuring all methods are constrained to exactly one additional gradient computation per step.
>
> > W3, Q2
>
> Our primary goal in Figure 1 is to evaluate generalization measures that are directly optimizable during the training process, rather than solely searching for the best post-hoc predictor. We agree that such a comparison is valuable for understanding predictive power.
>
> - **Focus on Single-Model Tractability**: While output-based metrics like disagreement and inconsistency (Johnson & Zhang, 2023) are excellent predictors, they require training and aggregating outputs from multiple models across different data splits. This multi-model dependency makes them computationally impractical to use as regularizers during training. Figure 1 intentionally compares local inconsistency against Hessian-based sharpness measures because they share the crucial practical advantage of being computable from a single model on a single data split.
>
> - **Bridging Prediction and Optimization**: The core significance of local inconsistency lies in successfully bringing the promising concept of "inconsistency" into the optimization realm. By designing a metric that is both differentiable and computable from a single trained model, we overcome the fundamental limitations of prior metrics and make it possible to directly regularize inconsistency (via IAM) within standard training pipelines.

---

> > ### Author Rebuttal · Reviewer_VTWQ · 2026-04-01
> >
> > Thank you very much for the careful rebuttal. Most of my concerns were addressed except Q1 and W2.
> >
> > The authors state that replacing the constraint $||\delta|| \le \rho$ with $||\delta|| = \rho$ is equivalent **“under the local quadratic expansion”**. However, this conditional equivalence is precisely what highlights that the method relies on an approximation whose validity depends on assumptions.
> >
> > My main concern remains that the paper’s claims are framed around the proposed exact local inconsistency measure, while all experiments are conducted using the approximated procedure in Algorithm 1. In particular, without either (i) a tractable instantiation of the exact method on smaller settings, or (ii) a quantitative analysis of the impact of these approximations, the connection between the theoretical objective and the observed empirical improvements remains under-specified. As a result, improvements demonstrated using Algorithm 1 do not directly validate the effectiveness of the underlying local inconsistency measure, but rather the effectiveness of Algorithm 1 itself.
> >
> > This point remains insufficiently addressed in the rebuttal and is important for strengthening the validity of the proposed approach.

---

> > > ### Author Response · Authors · 2026-04-07
> > >
> > > We sincerely thank the reviewer for the thoughtful follow-up.
> > >
> > > The proposed exact local inconsistency measure is **well-approximated** (with a relative error~$0.23\\%$) by Algorithm 1 as shown in the table below. As suggested by the reviewer, we tested a tractable instantiation of the exact method on a smaller setting ("Theory Exact Value") and qualitatively compare it with the estimates ("Numerical Exact Maximum"$ and "Algorithm 1 Estimate"). [https://anonymous.4open.science/r/IAM-48EB/training_inconsistency_comparison.pdf ]
> > >
> > > To directly provide the **tractable instantiation of the exact method on smaller settings** that you suggested, we explicitly evaluate the following metrics using the 3-layer MLP on a 2D Gaussian-mixture (Appendix C):
> > >
> > > - **Theory Exact value**: Computed $\frac{\rho^2}{2}\lambda_{\max}(F)$ by explicitly eigen-decompose $F\in \mathbb{R}^{5393 \times 5393}$.
> > >
> > > - **Numerical Exact Maximum**: The true local maximum $S_\rho^\ast$, found via an extensive Projected Gradient Ascent (PGD) in the direction $\delta^\ast$.
> > >
> > > - **Algorithm 1 Estimate**: Our proposed approximation $\hat{S_\rho}$, calculated in the direction $\delta_{alg1}$.
> > >
> > > | Evaluation Dimension | Comparison Metric | Result | Key Implication |
> > > |---|---|---|---|
> > > | Objective Value Error $(\downarrow)$| Mean Relative Error ($\frac{S_\rho^\ast - \hat{S_\rho}}{S_\rho^\ast}$) | $0.23\\%$ | The objective value error remains low during training. |
> > > | Directional Alignment $(\uparrow)$| Cosine Similarity ($\delta_{alg1}$ vs $v_1$) | 0.9960 ± 0.0004 | Algorithm 1 reliably converges to the principal direction, aligning tightly with the theoretical principal eigenvector. |
> > > | Update Directions $(\uparrow)$| Cosine Similarity ($\nabla_\theta L_{\text{IAM-D}}$) | 0.9995 | update derection derived from all three metrics exhibit alignment. |
> > >
> > > To demonstrate that Algorithm 1 ($\hat{S_\rho}$) tightly tracks the theoretical objective ($\frac{\rho^2}{2}\lambda_{\max}(F)$) with negligible error, we monitored the optimization dynamics throughout the entire training process [https://anonymous.4open.science/r/IAM-48EB/training_inconsistency_comparison.pdf ]. Every 10 steps, we computed the exact theoretical maximum via explicit eigen-decomposition and compared it against two values:
> > > - The numerical exact maximum ($S_\rho^\ast$): Computed using a extensive PGD optimizer (20 multi-starts, 200 steps, initial step size of $\rho/8$, with scheduled decay to guarantee fine-grained maxima detection).
> > > - Our proposed approximation ($\hat{S_\rho}$): Generated directly by Algorithm 1, which successfully captured $99.77\\%$ of the maximum value (a mean relative error of merely $0.23\\%$).
> > >
> > > As clearly illustrated in the provided figure, the trajectories of all three metrics virtually overlap throughout the entire training process, visually confirming that Algorithm 1 maintains a highly accurate approximation from start to finish.
> > >
> > > The training trajectory reveals a near-perfect alignment across three crucial evaluation dimensions:
> > >
> > > - Low Objective Value Error: The calculated objective value $\hat{S_\rho}$ consistently tracks both $S_\rho^\ast$ and $\frac{\rho^2}{2}\lambda_{\max}(F)$ across all stages of training. This yields an exceptionally tight mean relative error (calculated as $\frac{|S_\rho^\ast - \hat{S_\rho}|}{S_\rho^\ast}$) of merely $0.23\\%$.
> > > - High Directional Alignment ($\delta_{alg1}$ vs $v_1$): Algorithm 1 reliably converges to the principal direction. The path of our approximation $\delta_{alg1}$ aligns tightly with the theoretical principal eigenvector $v_1$, achieving a cosine similarity of 0.9960 ± 0.0004 (mean std).
> > > - Identical Update Directions ($\nabla_\theta L_{\text{IAM-D}}$): Crucially, when applied as a penalty for the IAM update, the penalty gradients ($\nabla_\theta$) derived from all three metrics ($\frac{\rho^2}{2}\lambda_{\max}(F)$, $S_\rho^\ast$, and $\hat{S_\rho}$) exhibit a continuous cosine similarity of 0.9995 throughout the entire training process.
> > >
> > > Together, these evaluations validate that Algorithm 1 accurately captures the intended worst-case local inconsistency—both in its value and its resulting optimization trajectory.

---

### Official Review · Reviewer_XUyg · 2026-03-10

**Soundness:** 3
**Presentation:** 3
**Significance:** 3
**Originality:** 3
**Overall Recommendation:** 4
**Confidence:** 3

**Summary:**

This paper proposes a local inconsistency generalization measure and a subsequent optimization algorithm with the inconsistency regularizer. The local inconsistency is defined as the maximum KL divergence of the model's output compared to a perturbed counterpart within a ball. This measure can be computed with a single model with unlabeled data. Empirical validation shows the proposed measure is effective in capturing the generalization gap, and the author theoretically connects it to the FIM and Hessian, establishing a generalization bound. Two optimization variants are proposed, and experiments across supervised, semi-, and unsupervised settings show improvements on vision tasks.

**Compliance With Llm Reviewing Policy:**

Affirmed.

**Final Justification:**

The author has addressed my concerns properly and demonstrated clearly that under semi- or self-supervised settings, the IAM has a clear advantage. Once their narrative on Figure 1 is corrected, my concerns are all addressed. Therefore, I'd maintain my positive assessment of the paper.

**Key Questions For Authors:**

See weaknesses.

**Limitations:**

Not thoroughly discussed.

**Strengths And Weaknesses:**

Strengths:
1. The proposed metric is technically novel and interesting. And its connection to FIM and generalization bounds provides a good theoretical guarantee.
2. The local inconsistency can be measured with a single model and does not require labeled data.
3. The IAM's performance improvement in semi-supervised and unsupervised tasks is strong.

Weaknesses:
1. For Fig. 1, the author demonstrates the advantage of local inconsistency over other sharpness metrics by showing how consistent this metric is over the whole family of models trained with different parameters, whereas sharpness metrics are only effective within each subgroup. But this comparison seems a little problematic for two reasons: 1) the cross-model, cross-hyperparameter correlation is less important when the proposed regularization works with only a single model; 2) it should not strike as a surprise that the absolute scale of sharpness (both trace and max eigenvalue of Hessian) isn't comparable across regimes induced by different training hypers. It is a valid argument for a model selection or evaluation metric. However, IAM is used as a within-run training regularizer, where these hyperparameters are fixed throughout training. The pathological cross-subgroup comparison never arises in practice.
2. The self-supervised evidence would be stronger with SimCLR results on CIFAR-100 as well, to test whether the gains persist on a more challenging dataset.
3. The clever symmetry argument used to cancel out the first-order term in expectation in the derivation of IAM-S is interesting, but neglects some practical concerns of iterative optimization. How large is the variance across steps, and how does this zero-mean per-step noise affect the convergence behavior of IAM-S?
4. A minor writing suggestion: many of the figures' captions are not self-contained and require some effort to understand.

---

> ### Author Rebuttal · Authors · 2026-03-31
>
> We thank the reviewer for highlighting our theoretical soundness, technical novelty, and strong empirical gains in label-scarce settings. To address your constructive feedback, the final revision will include an expanded Limitations section and fully self-contained figure captions, ensuring experimental setups and key takeaways are clear without cross-referencing the main text.
>
> ---
> > On Figure 1 and Cross-Regime Correlation (W1)
>
> Figure 1 is designed strictly for metric validation, not for simulating within-run optimization. Following standard methodology, it rigorously establishes local inconsistency ($S_ρ$) as a fundamental generalization metric capable of tracking the generalization gap across diverse models and hyperparameter regimes.
>
> For actual within-run training dynamics, Figure 3 directly addresses this point. It demonstrates that under a fixed setting, IAM-D actively suppresses the growth of $S_\rho$, mitigating overfitting and improving test accuracy over SGD. We will explicitly clarify the distinct purposes of these two figures in the revision.
>
> Please let us know if this addresses your concern, or if we have misunderstood any aspect of your question.
>
> ---
> > On SSL Evaluation on CIFAR-100 (W2)
>
> We have completed the SimCLR evaluation on CIFAR-100, confirming that IAM-D's effectiveness extends to more complex datasets. Please refer to this figure https://anonymous.4open.science/r/IAM-48EB/SimCLR_cifar100.pdf. It shows better convergence and linear probing accuracy.
>
> ---
> > W3 The clever symmetry argument used to cancel out the first-order term in expectation in the derivation of IAM-S is interesting, but neglects some practical concerns of iterative optimization. How large is the variance across steps, and how does this zero-mean per-step noise affect the convergence behavior of IAM-S?
>
> The symmetry argument in Sec. 5.1 is intended to justify cancellation of the first-order term in expectation, not to imply zero variance at each optimization step. More importantly, the purpose of this argument is to clarify the optimization bias of IAM-S: unlike a SAM-style interpretation that emphasizes the first-order effect around the loss gradient, IAM-S is designed so that the perturbed objective places greater emphasis on the curvature term
> $δ^⊤F(θ)δ$ (equivalently $δ^⊤G(θ)δ$ under the GN/FIM connection), i.e., on reducing local inconsistency along the principal eigenspace.
>
> At the same time, incomplete cancellation of the first-order term is not necessarily harmful. If we differentiate the first-order term $δ^⊤ ∇ L(θ)$ while detaching or locally freezing $δ$, the resulting contribution is approximately $Hδ$. From the perspective of edge-of-stability dynamics, such oscillatory Hessian-aligned components can induce a beneficial net drift toward lower sharpness over longer time scales, rather than acting as mere zero-mean noise (Cohen et al., 2021; Damian et al., 2023). This interpretation is consistent with recent analyses of SAM and EOS: Long and Bartlett (2024) shows that SAM exhibits oscillatory dynamics near high-curvature directions and that its update can be decomposed into a component associated with oscillation and a component that decreases the operator norm of the Hessian; related central-flow analyses likewise explain how oscillations can produce an implicit sharpness-reduction effect on the time-averaged optimization trajectory (Cohen et al., 2025).
>
> In practice, the step-wise variance is moderated by mini-batch/sub-batch gradient aggregation, and in distributed settings we expect independent perturbations across sub-batches to further average out the zero-mean component. Crucially, the perturbation is not merely arbitrary zero-mean random noise: Appendix C (Figs. 5–6) shows that even a single-step perturbation $δ_1$ meaningfully changes the decision boundary and is strongly aligned with the principal eigenspace of the FIM. We will clarify this intended interpretation of IAM-S more explicitly in the revision, and we are also running additional experiments to quantify the step-wise variance across seeds and its impact on convergence.
>
>
> ---
> **Reference**
>
> Cohen, J., Kaur, S., Li, Y., Kolter, J. Z., and Talwalkar,
> A. Gradient descent on neural networks typically occurs at the edge of stability. In International Conference on Learning Representations, 2021.
>
> Damian, A., Nichani, E., and Lee, J. D. Self-stabilization:
> The implicit bias of gradient descent at the edge of
> stability. In The Eleventh International Conference
> on Learning Representations, 2023.
>
> Long, P. M., & Bartlett, P. L. Sharpness-Aware Minimization and the Edge of Stability. Journal of Machine Learning Research (JMLR), 25(179), 1-20, 2024.
>
> Cohen, J., Damian, A., Talwalkar, A., Kolter, J. Z., and Lee,
> J. D. Understanding optimization in deep learning with
> central flows. In The Thirteenth International Conference
> on Learning Representations, 2025.

---

> > ### Author Rebuttal · Reviewer_XUyg · 2026-03-31
> >
> > Thank you for the comprehensive response.
> >
> > My concern about Figure 1 is mostly rhetorical. On the right column of line 242, the paper says "local inconsistency captures information about the generalization gap that is distinct from, or complementary to, traditional Hessian-based sharpness." My concern is that the negative overall correlations of Hessian-based sharpness are likely a result of the absolute scale difference introduced by variances in models and data augmentation. The positive correlation of local inconsistency with generalization gap likely comes from its output-based formulation being more invariant to absolute scale effects. So it does not necessarily capture complementary information compared to Hessian-based sharpness. One can reasonably argue it is more robust across different regimes, but the description in the manuscript is likely an overclaim.
> >
> > For W3, the EOS connection is genuinely interesting and, if developed properly, would strengthen the paper considerably.

---

> > > ### Author Response · Authors · 2026-04-08
> > >
> > > We thank the reviewer for this insightful clarification.
> > >
> > > As the reviewer suggested, we will refine the relevant statement to emphasizes the **robustness of the output-based formulation
> > > to absolute scale effects**.
> > >
> > > We agree with the reviewer's analysis: the output-based formulation of local
> > > inconsistency is more robust to absolute scale effects than Hessian-based
> > > sharpness, which likely explains its consistency across diverse training regimes.
> > >
> > > We believe this revision will result in a more accurate and stronger
> > > presentation of our contribution, and we appreciate the reviewer's careful
> > > reading that led to this improvement.
> > >
> > > We are glad the reviewer finds the EOS connection genuinely interesting. We
> > > agree that this connection deserves a more thorough treatment, and we will
> > > develop it more carefully in the revised manuscript — providing a clearer
> > > exposition of how the oscillatory Hessian-aligned components relate to
> > > edge-of-stability dynamics and their role in the implicit sharpness-reduction
> > > effect of IAM.

---

### Official Review · Reviewer_zypc · 2026-03-13

**Soundness:** 3
**Presentation:** 2
**Significance:** 1
**Originality:** 2
**Overall Recommendation:** 3
**Confidence:** 2

**Summary:**

The authors provide an alternative quantity they claim correlates with the generalization of the network called local inconsistensy, defined at every weight for the neural net as the maximum amount of kl divergence form the prior model by perturbing the model in the weight space. The claim two desirable properties of this measure: (1) it can be approximated by the largest eigenvalue of the fisher infromation matrix at every point, which can be approximated by Hessian of the loss, which enables them to obtain a generalization bound based on this quantity. (2) Unlike measures based on Hessian such as sharpness, it is label independent, hence can be computed more rigorously and independent of data.

**Compliance With Llm Reviewing Policy:**

Affirmed.

**Key Questions For Authors:**

can authors provide proof of their upper bound when they substitue the Hessian with the Gauss Newton matrix?

can the authors expain how their method differs form the Fisher Sam paper?

Is the gain you observe on CFAR-100 consistent over different seeds?

**Limitations:**

The presentation and comaprison to related work such as Fisher Sam is not done properly, so it is hard to see the novelty and contribution of the work if any.
Please see the weaknesses above.

**Strengths And Weaknesses:**

Strength:
The authors provide a new measure of generalization and they propose inconsistency measure aware minimization of the loss, as an alternative to sharpness aware minimization of the loss. they show some empirical improvements over shaprness minimization in some settings in their experiments.

Weakness:
The authors claim the fisher information matrix can be approximated well by the hessian matrix but they provide no proof/evidece. Their generalization bound is further based on this approixmation, i.e. they start from an well-known sharpness based generalization bound, and translate it to their measure using this approximation, which is not a novel theoretical result in my opinion.

Their experiments does not show a major improvement over sharpness aware minimization, which raises thei question of if there is any clear/consistent benefit of their method over sharpness aware minimization.

---

> ### Author Rebuttal · Authors · 2026-03-31
>
> We believe the main contribution of the paper may not have been fully reflected in the review.
>
>
> > Their experiments does not show a major improvement over sharpness aware minimization, which raises thei question of if there is any clear/consistent benefit of their method over sharpness aware minimization.
>
>
> **IAM offers a clear benefit over SAM**: our primary contribution—emphasized throughout the paper—is introducing a fundamentally **label-agnostic regularization metric**, beyond the simply aiming to improve SAM in standard supervised tasks.
>
> While SAM and its variants strictly require ground-truth labels to compute worst-case loss , our local inconsistency measure is evaluated entirely using unlabeled data. This unique property successfully unlocks principled, sharpness-aware regularization for **semi-supervised and self-supervised learning** frameworks (e.g., FixMatch, SimCLR), where label-dependent methods like SAM cannot be directly or effectively applied.
>
> ---
> > The authors claim the fisher information matrix can be approximated well by the hessian matrix but they provide no proof/evidece.
>
> **We explicitly provided the mathematical proof detailing the relationship between the Hessian and the Fisher Information Matrix (FIM) in Appendix A**.
>
> To clarify the direction of the approximation, our analysis proves that the **FIM approximates the Hessian**. We do not assume a blanket, unjustified replacement. Instead, Section 3.3 and Lemma A.1 rigorously establish that for softmax cross-entropy, the empirical Hessian admits an exact decomposition into a Gauss-Newton (empirical FIM) term plus a residual term ($H=F+R$). Under the near-interpolation regime typical in modern deep learning, this residual term is demonstrably bounded and becomes negligible, which mathematically justifies the FIM as a sound approximation of the Hessian.
>
> $$∇\_θ^2 l_i=\underbrace{∇\_θ z\_i^⊤ (∇\_z^2 l_i) ∇\_θ z\_i}\_{\text{GN Term}}+\underbrace{\sum\_{k=1}^C (f(x\_i;θ)-y\_i)_k \cdot ∇\_θ^2 z\_k(x\_i;θ)}\_{\text{Residual Term}},$$
> and thus
> $$H_S=F_S+R_S.$$
>
> > can authors provide proof of their upper bound when they substitue the Hessian with the Gauss Newton matrix?
>
> Our theoretical claim is therefore more precise than a huristic $H(θ)=F(θ)$ substitution. The GN term coincides with the empirical FIM term for softmax cross-entropy, while the gap between the Hessian and this term is captured by the residual $R_S(θ)$. Appendix A then uses Weyl’s inequality to show
>
> $$λ_{max}(H_S)-λ_{max}(F_S)\le ||R_S||_2.$$
>
> In modern overparameterized deep learning, models are often trained to a near-zero training error. At such a minimum, residual ($f(x_i;θ)-y_i$) is correspondingly small, which justifies replacing the Hessian term in the bound by the FIM term up to a controlled slack, rather than by an unconditional exact substitution. Specifically, Papyan (2018) demonstrated that the dominant outliers originate entirely from the $G$ term (which is equivalent to the FIM, $F$), while the $R$ term contributes only to the bulk. Therefore, our derivation $H_S = F_S + R_S$ strictly follows this standard, textbook decomposition, fully supporting our generalization bound.
>
>
> ---
> > fundamental distinction between IAM and Fisher-SAM
>
> The fundamental distinction between IAM and Fisher-SAM is that **IAM is inherently label-agnostic**. While Fisher SAM seeks a parameter region robust to worst-case supervised loss, IAM minimizes worst-case output inconsistency. This conceptual shift allows IAM to utilize vast amounts of unlabeled data, making it directly applicable as a plug-and-play regularizer in Semi-Supervised Learning (SSL) and Self-Supervised Learning frameworks like FixMatch and SimCLR which is not possible for standard SAM variants.
>
> |Feature|FisherSAM|IAM (Ours)|
> |-|-|-|
> |Maximization Objective|Loss: $l(θ+ϵ)$|**Output Divergence**: $KL(f(θ)\|\|f(θ+δ))$|
> |Neighborhood Constraint|FIM Ellipsoid: $ϵ^⊤F(θ)ϵ \le γ^{2}$ | Euclidean Ball: $\|δ\|_2\le ρ$ |
> |Perturbation Direction|$\hat{F}^{-1}∇ l(θ)$|$F(θ)\varepsilon$ (K=1)|
> |Label Requirement|Required|**Not Required**|
>
> As the table illustrates, IAM introduces a novel objective that targets the stability of the model's output distribution. This conceptual shift yields a perturbation direction distinct from Fisher SAM’s preconditioned loss ascent, and crucially, provides the foundation for its performance in label-scarce environments.
>
> > Is the gain you observe on CFAR-100 consistent over different seeds?
>
> Yes. The CIFAR-100 result reported in Table 1 is the mean ± stderr across runs with random seed: SAM 17.63 ± 0.12 vs IAM-S 16.82 ± 0.01, indicating the gain is consistent across seeds.

---

### Decision · Program_Chairs · 2026-04-30

**Decision:**

Accept (regular)

**Comment:**

The paper proposed IAM, a label-agnostic inconsistency-based regularization framework for improving generalization. Most reviewers agreed that the main idea is interesting and that the ability to compute the regularizer without labels is a real strength, especially for semi-supervised and self-supervised learning. The main concerns were about the theoretical justification, particularly the connection to the Fisher information matrix and Hessian, the role of the one-step approximation used in practice, and whether the empirical gains in standard supervised settings are large enough. The rebuttal addressed these concerns reasonably well. The authors clarified the difference from Fisher-SAM, added a comparison between the exact objective and the practical approximation, and strengthened the discussion of why the method is most useful in label-scarce settings compared to SAM in standard supervised learning. While the theory is not entirely novel, the contribution is meaningful, the empirical results are strong, and the main concerns were sufficiently addressed. Overall, I believe that this paper makes a solid contribution and should be accepted.